# Reliable Thinking with Images

**Haobin Li**[1]  **Yutong Yang**[1]  **Yijie Lin**[1]  **Xiang Dai**[2]  **Mouxing Yang**[†1]  **Xi Peng**[†34]

https://github.com/XLearning-SCU/Reliable_TWI

## Abstract

As a multimodal extension of Chain-of-Thought (CoT), Thinking with Images (TWI) has emerged as a promising avenue to enhance the reasoning capability of Multi-modal Large Language Models (MLLMs), which generates interleaved CoT by incorporating visual cues into the textual reasoning process. However, the success of existing TWI methods heavily relies on the assumption that interleaved CoTs are faultless, which is easily violated due to the complexity of multimodal understanding. In this paper, we reveal and study a highly-practical yet under-explored problem in TWI, termed Noisy Thinking (NT). Specifically, NT refers to the imperfect visual cues mining and answer reasoning process. As the saying goes, "One mistake leads to another", erroneous interleaved CoT would cause error accumulation, thus significantly degrading the performance. To solve NT, we propose a novel method dubbed Reliable Thinking with Images (RTWI). In brief, RTWI estimates the reliability of visual cues and textual CoT in a unified text-centric manner and then employs robust filtering and voting modules to prevent NT from contaminating final answer. Extensive experiments on seven benchmarks verify the effectiveness of RTWI against NT.

## 1. Introduction

*The devil is in the details.*

*Ludwig Mies van der Rohe*

[1]College of Computer Science, Sichuan University [2]Southwest China Institute of Electronic Technology [3]School of Artificial Intelligence, Sichuan University [4]Tianfu Jincheng Laboratory, Chengdu, China. Correspondence to: Mouxing Yang <yang-mouxing@gmail.com>, Xi Peng <pengx.gm@gmail.com>.

*Proceedings of the 43rd International Conference on Machine Learning*, Seoul, South Korea. PMLR 306, 2026. Copyright 2026 by the author(s).

Chain-of-Thought (CoT) (Wei et al., 2022; Zhang et al., 2022) has become the dominant paradigm for reasoning with Large Language Model (LLM) (Naveed et al., 2025; Zhao et al., 2023), which solves complex problems by generating a sequence of textual reasoning steps. Inspired by this, recent advances in Multimodal Large Language Model (MLLM) (Zhang et al., 2023; Zheng et al., 2023) primarily adopt the "reasoning within language" paradigm over textual and visual inputs to enhance multimodal reasoning capability. However, such a paradigm simply encodes the whole visual input as the static context, making it difficult to exploit task-relevant visual information. In other words, visual inputs are always rich and redundant, which might obscure the critical details necessary for accurate answer derivation. As a remedy, endowing MLLM with the capability to "thinking with images" (OpenAI, 2025; Zheng et al., 2025; Lai et al., 2025) has recently emerged as a promising paradigm, which actively manipulates visual information in intermediate steps and thus forms interleaved image-text CoTs. Specifically, Thinking With Images (TWI) paradigm typically consists of two stages: i) **Cue Mining**: MLLMs inspect the whole visual inputs and generate textual CoT to call external tools for image manipulation, thereby acquiring task-relevant visual cues; ii) **Answer Reasoning**: with the desirable visual cues, MLLMs have incentives to derive the correct answer and solve the task.

Despite the promising performance of the existing TWI approaches, their success heavily relies on the assumption that both the cue mining and answer reasoning stages are faultless. However, in the real-world TWI applications, such an ideal assumption is daunting and even impossible to satisfy, leading to the **Noisy Thinking (NT)** problem. Specifically, as shown in Fig. 1(a), NT refers to the noisy cue mining and noisy answer reasoning in the TWI process. Specifically, i) **Noisy Mining**: akin to finding needles in a haystack, it is inevitable to extract task-irrelevant or coarse-grained cues from redundant visual inputs; ii) **Noisy Reasoning**: even with desirable visual cues, the inherent limitations of MLLMs in multimodal understanding would contaminate the textual CoT and lead to incorrect answer. As the saying goes, "One mistake leads to another", either the undesirable visual cues in the mining stage or the erroneous textual CoT in the reasoning stage would propagate along the TWI

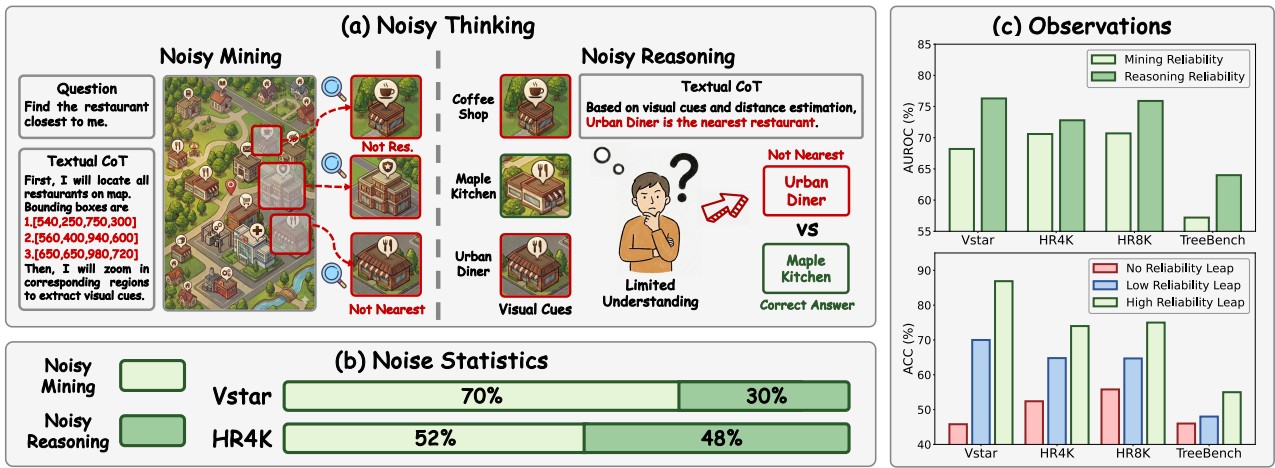

*Figure 1.* (a) **Noisy Thinking:** the TWI paradigm would inevitably suffer from the noisy thinking problem at both the mining and reasoning stages. On the one hand, MLLMs would extract task-agnostic or inaccurate visual cues once the textual CoT generates unreliable tool-calling instructions. Clearly, it is difficult to derive the right answer with incorrect visual cues, *e.g.*, the question is "Find the restaurant closest to me.", but the visual cues either do not depict restaurants or correspond to the non-closest one. On the other hand, even desirable information exists in acquired visual cues, MLLM might generate erroneous textual CoT and then the incorrect answer, *e.g.*, failing to identify the nearest restaurant due to limited distance estimation. (b) **Noise Statistics:** we investigate the TWI cases in the Vstar and HR4k datasets and observe that NT in either mining or reasoning would lead to wrong answers in real-world scenarios. (c) **Observations:** we estimate the reliability of the two stages in the TWI process using tool-invocation and reasoning tokens, and then adopt the AUROC metric to quantify the relationship between reliability and reasoning accuracy. As a result, one could observe that traces with correct answers tend to exhibit higher dual-stage reliabilities and reliability leap from mining stage to reasoning stage.

process, thus significantly degrading the performance.

To handle the NT problem, the most promising solution might be the Test-Time Scaling (TTS) (Zhang et al., 2025a; Muennighoff et al., 2025), which improves reasoning accuracy by sampling multiple reasoning paths and deriving answers via search- (Yan et al., 2025; Xie et al., 2023) or consistency-based (Wang et al., 2022; Kang et al., 2025) mechanisms. However, most existing TTS methods are specifically designed for "reasoning with language" paradigm in LLM, while overlooking the complexity of "thinking with images" paradigm in MLLM. Specifically, they are intractable for addressing the NT problem due to the following two reasons. On the one hand, unlike discrete and tokenized text, visual signals are continuous and unsegmented, making visual uncertainty estimation substantially more challenging than token-level uncertainty modeling. Moreover, visual uncertainty is not necessarily aligned with cue uncertainty, as it might indicate image quality yet fail to reveal whether the mined cue is task-relevant or not. On the other hand, it remains under-explored how inaccurate visual cues propagate into and distort subsequent textual reasoning, not to mention how to achieve NT-robust TTS.

Instead of exhaustingly modeling visual uncertainty, we argue that the image-manipulation textual CoT itself could implicitly serve as a natural proxy for visual cue reliability, thus enabling a dual-stage robust TTS framework to tackle NT in the TWI process. To support our claims, we present two observations in Fig. 1(c). To be specific, i) Reliability

Correlation: both the reliabilities of the visual cues in the mining stage and the textual CoT in the reasoning stage are correlated with final performance, *i.e.*, reliable visual cues and answer reasoning steps would facilitate the obtaining of correct answers; ii) Reliability Leap: the correct visual cues tend to induce more reliability leap from the mining to the reasoning stage. Intuitively, akin to human cognition, the acquisition of correct visual cues would boost the confidence of the subsequent reasoning process.

Based on the above discussions and observations, we propose a TTS-based TWI method to achieve robustness against NT, dubbed Reliable Thinking With Images (RTWI), which samples multiple thinking traces for mining and deriving trustworthy answer. In brief, RTWI adopts a text-centric reliability estimation mechanism, which could identify noisy mining and reasoning in a unified manner. With formulated reliability, RTWI employs a dual-stage filtering module to discard unreliable thinking traces with self-adaptive thresholds, thereby preventing NT from contaminating answer derivation. Finally, RTWI conducts NT-robust voting over the remaining traces by resorting to the reliability leap characteristic, which contributes to deriving trustworthy answer. In summary, the major contributions are given as follows.

- We reveal and study a novel and practical problem in TWI, termed Noisy Thinking (NT). In brief, NT refers to the noisy mining and reasoning stages in the TWI process, which would propagate errors through the

subsequent reasoning and significantly degrade MLLM performance.

- To achieve robust TWI against NT, we propose a novel method termed RTWI. In brief, RTWI estimates the reliability of visual cues and textual CoT in a unified text-centric manner and then mitigates the negative impact of NT through dual-stage filtering and voting.

- The proposed RTWI could not only mitigate NT on offline-generated traces, but also enable reliable early stopping during online generation and thus significantly improve efficiency. Extensive experiments on seven benchmarks under offline and online settings demonstrate the effectiveness of RTWI against noisy thinking compared with state-of-the-art methods.

## 2. Related Work

In this section, we provide a brief review of two topics highly related to this work, including thinking with images and test-time scaling.

### 2.1. Thinking with Images

Thinking with images aims to solve complex multimodal questions by actively manipulating images (*e.g.*, zooming into regions of interest) to support subsequent reasoning, which is popularized by OpenAI o3 (OpenAI, 2025). From the perspective of how to endow MLLMs with the TWI capability, existing approaches could be categorized into two groups: i) prompt-based methods (Li et al., 2025; Shen et al., 2025), which carefully design prompts to coordinate external tools for extracting visual cues without updating model parameters; ii) training-based methods (Zhang et al., 2025b; Zheng et al., 2025; Lai et al., 2025), which conduct supervised fine-tuning on predefined trajectories to learn structured reasoning patterns, or adopt reinforcement learning with outcome-based rewards to explore visual manipulation and reasoning policies autonomously.

Among existing approaches, DRIM (Yang et al., 2025b) is the most relevant to our work. In brief, DRIM leverages a carefully-designed reinforcement learning scheme to strengthen the self-reflection capacity of MLLMs during the TWI process. In contrast, rather than modifying training to improve robustness, we propose a reliable TTS mechanism at inference time, which is orthogonal to DRIM and could serve as a plug-and-play solution for a wide spectrum of TWI models, enabling more robust thinking.

### 2.2. Test-time Scaling

Test-time Scaling has become the dominant paradigm to improve reasoning accuracy in LLM, revealing a scaling law that more compute leads to better performance, as supported by representative models include OpenAI o1 (Jaech et al., 2024), DeepSeek R1 (Guo et al., 2025), Grok-4 (xAI, 2025), and Qwen3 (Yang et al., 2025a). Based on the ways to utilize extra test-time compute, the existing TTS approaches could be divided into three categories: i) self-consistency methods (Wang et al., 2022), which sample multiple reasoning trajectories and aggregate them via majority voting; ii) early stopping methods (Li et al., 2024; Aggarwal et al., 2023), which seek better accuracy-compute trade-offs by adaptively terminating further token generation once the existing traces reach consensus; iii) robust reasoning methods (Fu et al., 2025; Kang et al., 2025), which dynamically filter out low-confidence traces or scale low-confidence reasoning steps to improve reasoning accuracy.

The major differences between existing TTS methods and this work are given below. On the one hand, existing TTS methods are carefully designed for textual CoT in "reasoning with language" paradigm, whereas our approach addresses the more challenging interleaved CoT in "thinking with images" paradigm. On the other hand, to the best of our knowledge, no prior work has explored how to identify undesirable visual cues, not to mention mitigating the negative impacts of NT in the TWI process.

## 3. Method

In this section, we introduce Reliable Thinking With Images (RTWI) to tackle NT for TWI. In Section 3.1, we present the formal definition of the TWI process. In Section 3.2, we propose a novel reliability estimation mechanism to identify NT. In Section 3.3-3.4, we introduce dual-stage filtering and reliable voting modules to achieve robustness against NT.

### 3.1. Problem Formulation

Without loss of generality, we take the one thinking trace $t$ as a showcase to elaborate on the TWI process. Let $\mathcal{V}$ denotes the space of all possible textual outputs (*e.g.*, the vocabulary of text tokens), $\mathcal{I}$ denotes the space of all possible intermediate visual cues, and $\Theta$ represents the parameters of the MLLM. Given a question $Q$ with an initial image $I$, TWI aims to derive the answer by generating a sequence of interleaved image-text CoT $t^m = \left\{ (\mathbf{x}_i^{\text{txt}}, \mathbf{x}_i^{\text{vis}}) \right\}_{i=1}^{N-1}$ in the mining stage and subsequently facilitating the textual CoT $t^r = \{x_N^{\text{txt}}\}$ in the reasoning stage, where $N$ denotes the total number of turns, $\mathbf{x}_i^{\text{txt}} \in \mathcal{V}$ and $\mathbf{x}_i^{\text{vis}} \in \mathcal{I}$ denote the textual CoT and the corresponding visual cues at $i$-th turn, respectively. Specifically,

**Mining Stage ($i < N$):** the interleaved image-text CoT could be derived as

$$
\begin{aligned}
\mathbf{x}_i^{\text{txt}} &\sim P(\,\cdot \mid S_i, I, Q; \Theta), \\
\mathbf{x}_i^{\text{vis}} &= f_{\text{tool}}(\mathbf{x}_i^{\text{txt}}, I),
\end{aligned}
\tag{1}
$$

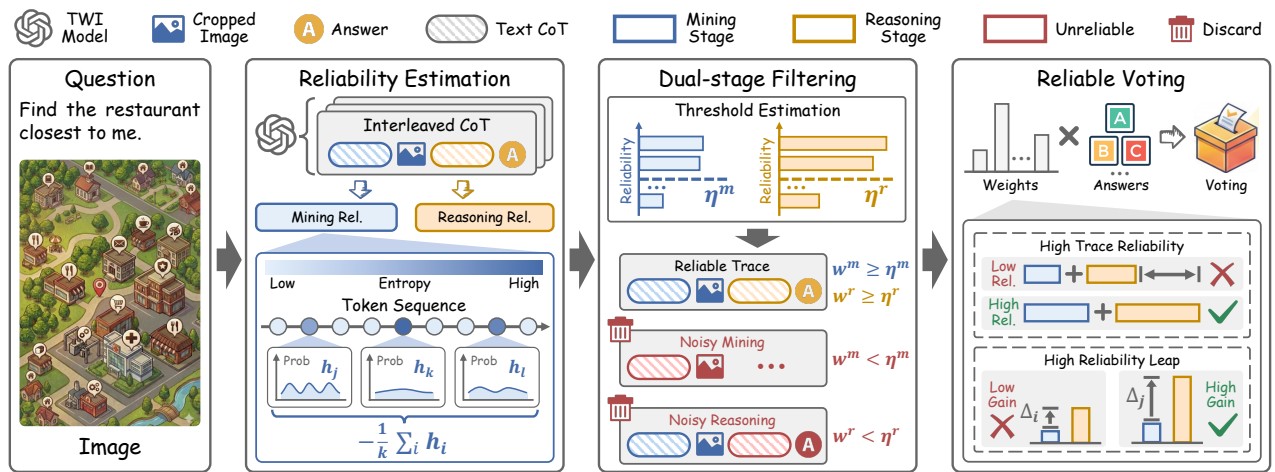

*Figure 2.* Overview of our method RTWI. For clarity, we take single-turn cue mining as a showcase and denote stage reliability $w(t^s)$ as $w^s$ with $s \in \{m, r\}$. Given a multimodal question, RTWI first generates multiple interleaved traces and estimates the reliability of visual cues in the mining stage and textual CoT in the reasoning stage. After that, RTWI identifies and filters the unreliable traces with self-adaptive thresholds. Finally, RTWI assigns higher weights to trustworthy traces based on two carefully-designed principles and then aggregates them to derive the answer.

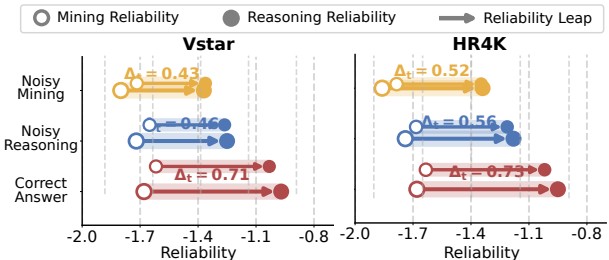

*Figure 3.* Failure case analyses. "Noisy Mining" and "Noisy Reasoning" indicate the underlying causes of incorrect answers.

where $S_i = \left\{ (\mathbf{x}_j^{\text{txt}}, \mathbf{x}_j^{\text{vis}}) \right\}_{j=1}^{i-1}$ denotes thinking history at $i$-th turn, $f_{\text{tool}}$ denotes the tool invocation (*e.g.*, zooming in), $P(\cdot \mid \cdot; \Theta)$ denotes the conditional distributions parameterized by MLLM, and $\sim$ indicates sampling process.

**Reasoning Stage** ($i = N$): based on the extracted visual cues, the MLLM generates textual CoT to conduct comprehensive reasoning and derive the answer,

$$\mathbf{x}_N^{\text{txt}} \sim P(\cdot \mid S_N, I, Q; \Theta),$$
$$A_t = f_{\text{ans}}(\mathbf{x}_N^{\text{txt}}), \tag{2}$$

where $A_t$ indicates the answer of trace $t$, $f_{\text{ans}}(\cdot)$ denotes the deterministic function that extracts the answer from the textual outputs. As discussed in Introduction, existing TWI methods assume that both the mining and reasoning stages are faultless, which is often violated in real-world scenarios and thus leads to the NT problem.

To tackle the NT problem, we propose Reliable Thinking With Images (RTWI) as shown in Fig. 2, which consists of the reliability estimation module, the dually robust filtering

module, and the reliable fusion module. In the following, we will elaborate on each of them.

### 3.2. Reliability Estimation

Recent studies (Farquhar et al., 2024; Taubenfeld et al., 2025) have demonstrated that the quality of thinking trace could be estimated from model-intrinsic signals without requiring external supervision, *e.g.*, token confidence and trace distribution. However, existing methods are carefully designed for "reasoning with language" paradigm, leaving an urgent need to identify both noisy visual cues and textual CoT as mentioned in Introduction.

To remedy this, instead of exhaustingly estimating visual uncertainty, we delve into the cue mining stage and reveal that the acquisition of visual cues is inherently reflected by the textual CoT. In other words, reliable textual CoT for tool invocation could act as the precondition for desirable visual cues derivation. Inspired by this, we propose a text-centric paradigm to estimate reliabilities of both visual cues and textual CoT, which facilitates the identification of NT. Specifically, we give a formal definition of stage reliability.

**Definition 3.1** (Stage Reliability). For a given trace $t$, the reliability of stage $t^s (s \in \{m, r\})$ is defined as

$$w(t^s) = -\frac{1}{k} \sum_{i \in \mathcal{K}(\mathcal{H}(t^s))} h_i^s, \tag{3}$$

where $\mathcal{H}(t^s)$ denotes the set of token entropies in $t^s$, $\mathcal{K}(\cdot)$ returns the indices of the Top-$k$ largest values, $w(t^s) \leq 0$ and $|w(t^s)|$ is negatively-correlated with the stage reliability.

More specifically, the entropy of $i$-th token in $\mathcal{H}(t^s)$ is de-

fined as $h_i^s = - \sum_{j \in \mathcal{V}} p_{ij}^s \log p_{ij}^s$, where $\{p_{ij}^s\}_{j \in \mathcal{V}}$ denotes the predicted probability distribution produced by MLLM. Such a reliability formulation embraces the following merits: i) it avoids considering unimportant or redundant tokens by concentrating on pivotal decision points with higher entropies; ii) the Top-$k$ selection yields more stable reliability estimation across mining and reasoning stages, regardless of CoT length (see more details in Appendix C.5). Intuitively, the unreliable textual CoT tend to exhibit low reliability, which could serve as a promising principle to identify NT.

To support our claims, we analyze the stage-wise reliabilities of the traces with either wrong or correct answers. From the results in Fig. 3, one could observe that the acquisition of undesired cues in the mining stage would propagate errors to the subsequent reasoning stage, leading to reliability degradation of the reasoning stage compared to correct traces. In contrast, traces with correct answers exhibit high reliability across both stages, further verifying that reliability could serve as a principle for identifying NT.

### 3.3. Dual-stage Filtering

To prevent unreliable traces from contaminating the answer, we propose to generate multiple thinking traces and prioritize trustworthy traces among them. Specifically, the reliable thinking traces could be determined as follows,

$$T_{\text{rel}} = \{ t \mid w(t^m) \geq \eta^m \ \wedge \ w(t^r) \geq \eta^r \}, \qquad (4)$$

where $T_{\text{rel}}$ denotes the set of reliable traces, $\eta^m$ and $\eta^r$ indicate the filtering thresholds in mining and reasoning stages, respectively. Specifically, for each multimodal question, we sample warm-up traces to derive stage-wise thresholds, i.e.,

$$\eta^s = \text{Percentile}_\alpha \left( \{ w(t^s) \mid t \in T_{\text{sel}} \} \right), \qquad (5)$$

where $s \in \{m, r\}$, $T_{\text{sel}}$ represents selected traces for threshold estimation, $\text{Percentile}_\alpha(\cdot)$ indicates that traces with the lowest $\alpha$ fraction of reliabilities are filtered at this stage. Such behavior could exclude unreliable traces with potentially erroneous mining or reasoning, thus facilitating the derivation of the correct answer. It is worth noting that the dual-stage filtering module could be easily integrated into existing TWI models under both online and offline settings, see more details in Alg. 1-2. More specifically, in the online pipeline, traces are generated one by one, and $T_{\text{sel}}$ denotes the subset of warm-up traces used to estimate the stopping thresholds, which enable dynamic early stopping for subsequent unreliable traces within the budget. In contrast, in the offline pipeline, all traces are pre-generated and $T_{\text{sel}}$ refers to the full set of generated traces used to estimate thresholds for post-hoc filtering.

### 3.4. Reliable Voting

Although the dual-stage filtering in Eq. 4 could exclude unreliable traces with NT to a great extent, it is still inadequate for determining trustworthy answer among the remaining ones. One straightforward solution is to aggregate the answers via majority voting (Wang et al., 2022), i.e., select the most common answer. However, such a simple approach treats each trace and the corresponding answer equally, regardless of their quality. To remedy this, we propose a novel reliable voting module to derive the trustworthy answer $\hat{A}$ as follows,

$$\hat{A} = \arg\max_a \sum_{t \in T_{\text{rel}}} C_t \cdot \mathbb{I}\big[A_t = a\big],$$
$$C_t = exp\left(\Delta_t / \left(|w_t| \cdot \tau\right)\right), \qquad (6)$$

where $w_t$ denotes the trace-wise reliability, $\Delta_t$ denotes the reliability leap, $\tau$ is the temperature, $\mathbb{I}[\cdot]$ is an indicator function evaluating to 1 *i.f.f.* the condition is satisfied. In particular, the confidence weight $C_t$ is determined by the dedicated two-fold principle. Inspired by the observations in Fig. 1(c) and Fig. 3, the traces with correct answers always exhibit higher dual-stage reliability and reliability leap, which could serve as two principles to determine the trustworthy answers. Formally, we give the following definition of the trace reliability.

**Definition 3.2** (Trace Reliability). For a given thinking trace $t$, the trace-wise reliability is defined as

$$w_t = \sum_{s \in \{m, r\}} w(t^s). \qquad (7)$$

Such a formulation encourages traces with reliable mining and reasoning stages to yield higher confidence. Moreover, we introduce the definition of reliability leap as follows.

**Definition 3.3** (Reliability Leap). For a given two-stage thinking trace $t = \{t^m, t^r\}$, the reliability leap is defined as

$$\Delta_t = \max\left(w(t^r) - w(t^m), 0\right), \qquad (8)$$

where $\max(\cdot, 0)$ ensures the reliability leap is non-negative.

The design of $\Delta_t$ could mine the potential correct answer by prioritizing traces with substantial reliability leap. In other words, beyond traces that are reliable in both stages, some traces may exhibit low reliability during the mining stage due to ambiguous multimodal understanding, yet still generate confident textual CoT and ultimately arrive at the correct answer after incorporating appropriate visual cues.

## 4. Experiments

In this section, we conduct extensive experiments on widely-used TWI benchmarks to verify the effectiveness of the proposed RTWI. Due to space limitation, we present more experimental details and results in Appendix C.

*Table 1.* Results on real-world high-resolution benchmarks under online setting. "ACC" and "TSR" denote the final accuracy and token saving ratio, respectively. The best results are marked in **bold**.

| | Vstar Bench | | | | HR-Bench 4K | | | | HR-Bench 8K | | | |
| | Attr | | Spatial | | FSP | | FCP | | FSP | | FCP | |
| Method | ACC | TSR | ACC | TSR | ACC | TSR | ACC | TSR | ACC | TSR | ACC | TSR |
|---|---|---|---|---|---|---|---|---|---|---|---|---|
| GPT-4o | 72.2 | - | 60.5 | - | 66.8 | - | 63.3 | - | 60.8 | - | 58.5 | - |
| Thyme | 83.5 | - | 80.3 | - | 91.0 | - | 63.0 | - | 86.5 | - | 57.5 | - |
| DeepEyes | 91.3 | - | 88.2 | - | 91.3 | - | 59.0 | - | 86.5 | - | 58.5 | - |
| *Qwen3-VL Thinking* | | | | | | | | | | | | |
| Base | 78.3 | - | 73.7 | - | 83.0 | - | 57.3 | - | 78.3 | - | 56.0 | |
| SC | 80.0 | - | 77.6 | - | 89.5 | - | 61.0 | - | 86.5 | - | 61.0 | - |
| ASC | 80.0 | 51.8 | 79.0 | 42.8 | 89.5 | 51.9 | 61.8 | 29.3 | 86.5 | 50.4 | 60.8 | 28.6 |
| ESC | 80.0 | 35.9 | 77.6 | 22.0 | 89.5 | 40.5 | 61.0 | 18.9 | 86.5 | 35.7 | 61.0 | 18.5 |
| CISC | 82.6 | 41.1 | 69.7 | 29.9 | 88.0 | 32.3 | 59.0 | 42.9 | 87.8 | 42.6 | 56.0 | 38.8 |
| DeepConf | 81.7 | 47.6 | 76.3 | 52.8 | 90.5 | 57.3 | 62.5 | 48.9 | 87.0 | 46.9 | 61.8 | 51.1 |
| Self-Cer. | 82.6 | 42.9 | 68.4 | 33.1 | 89.5 | 47.6 | 59.0 | 45.4 | 88.0 | 43.8 | 59.0 | 43.5 |
| Ours | **85.2** | **61.1** | **81.6** | **61.2** | **91.3** | **61.9** | **65.8** | **50.5** | **88.8** | **59.3** | **62.3** | **52.3** |
| *Qwen3-VL Instruct* | | | | | | | | | | | | |
| Base | 90.4 | - | 86.8 | - | 93.8 | - | 72.0 | - | 90.8 | - | 71.3 | |
| SC | 92.2 | - | 89.5 | - | 95.3 | - | 77.0 | - | 93.0 | - | 74.8 | - |
| ASC | 93.0 | 59.4 | 89.5 | 56.8 | 95.8 | 59.8 | 76.8 | 39.9 | 93.0 | 54.6 | 74.8 | 38.2 |
| ESC | 92.2 | 56.1 | 89.5 | 50.9 | 95.3 | 60.7 | 77.0 | 23.1 | 93.0 | 51.7 | 74.8 | 25.6 |
| CISC | 93.9 | 55.8 | 89.5 | 56.6 | 95.5 | 57.7 | 75.8 | 41.2 | 91.0 | 53.6 | 75.8 | 30.0 |
| DeepConf | 93.9 | 57.3 | **90.8** | 55.0 | 96.0 | 60.3 | 76.8 | 38.3 | 93.5 | 58.9 | 75.5 | 32.1 |
| Self-Cer. | 92.2 | 59.6 | 89.5 | 57.0 | 95.3 | 62.5 | 75.8 | 43.2 | 88.8 | 57.1 | 75.3 | 30.8 |
| Ours | **94.8** | **60.4** | **90.8** | **58.5** | **96.5** | **66.2** | **78.3** | **46.4** | **94.8** | **60.3** | **77.0** | **45.8** |

*Table 2.* Results on TWI-oriented TreeBench under online setting.

| | Reasoning | | | Perception | | |
| Method | ACC | mIoU | TSR | ACC | mIoU | TSR |
|---|---|---|---|---|---|---|
| *Qwen3-VL Thinking* | | | | | | |
| Base | 32.8 | - | - | 63.1 | - | - |
| SC | 30.9 | 37.8 | - | 67.1 | 32.5 | - |
| ASC | 30.9 | 37.7 | 32.1 | 67.1 | 32.1 | 40.1 |
| ESC | 30.9 | 37.7 | 10.7 | 67.1 | 33.4 | 28.0 |
| CISC | 27.7 | 36.5 | 48.9 | 67.1 | 32.9 | 49.7 |
| DeepConf | 35.6 | 39.3 | 51.7 | 65.8 | 31.9 | 53.5 |
| Self-Cer. | 30.9 | 37.6 | 32.3 | 64.4 | 33.1 | 38.9 |
| Ours | **37.1** | **40.8** | **59.4** | **67.8** | **35.1** | **60.1** |
| *Qwen3-VL Instruct* | | | | | | |
| Base | 34.0 | - | - | 64.4 | - | - |
| SC | 34.0 | 43.4 | - | 68.4 | 38.6 | - |
| ASC | 34.0 | 43.6 | 41.2 | 68.4 | 38.9 | 47.9 |
| ESC | 34.0 | 43.5 | 26.8 | 68.4 | 38.7 | 40.9 |
| CISC | 34.4 | 43.5 | 38.6 | 67.8 | 37.9 | 43.2 |
| DeepConf | 34.4 | 43.3 | 33.8 | 68.4 | 38.3 | 43.4 |
| Self-Cer. | 34.4 | 43.4 | 42.3 | 68.4 | 38.9 | 48.5 |
| Ours | **34.8** | **45.8** | **42.8** | **70.5** | **40.8** | **50.7** |

*Table 3.* Results on reasoning benchmarks under online setting.

| | MathVision | | LogicVista | |
| Method | ACC | TSR | ACC | TSR |
|---|---|---|---|---|
| *Qwen3-VL Thinking* | | | | |
| Base | 18.7 | - | 44.3 | - |
| SC | 19.9 | - | 46.4 | - |
| ASC | 19.9 | 8.6 | 46.4 | 21.2 |
| ESC | 19.9 | 5.5 | 44.3 | 18.9 |
| CISC | 20.1 | 16.1 | 49.8 | 30.8 |
| DeepConf | 21.2 | 33.3 | 51.8 | 34.5 |
| Self-Cer. | 18.0 | 15.1 | 49.3 | 31.7 |
| Ours | **23.2** | **34.6** | **61.7** | **50.5** |
| *Qwen3-VL Instruct* | | | | |
| Base | 21.4 | - | 47.9 | - |
| SC | 24.0 | - | 54.6 | - |
| ASC | 24.0 | 17.2 | 54.1 | 27.6 |
| ESC | 24.0 | 3.8 | 54.6 | 7.9 |
| CISC | 22.4 | 8.1 | 53.4 | 14.1 |
| DeepConf | 22.7 | 35.7 | 56.0 | 32.1 |
| Self-Cer. | 24.0 | 7.7 | 54.8 | 15.1 |
| Ours | **26.1** | **37.6** | **58.5** | **34.7** |

## 4.1. Experiment Configurations

**Settings.** Following (Li et al., 2024; Fu et al., 2025), we adopt two widely-used evaluation protocols, namely, online and offline settings. In the online setting, we perform on-the-fly token generation with up to 32 thinking traces per question, evaluating whether the methods could improve both reasoning accuracy and computational efficiency. In the offline setting, 32 thinking traces are pre-generated for each problem, and the key challenge is deriving trustworthy answer from multiple traces. Note that the number of warm-up traces $|T_{sel}|$ in the offline and online setting are set to 32

and 8, respectively.

**Benchmarks.** To comprehensively evaluate the effectiveness of RTWI, we evaluate our method on real-world high-resolution benchmarks, including Vstar Bench (Wu & Xie, 2024), HR-Bench (Wang et al., 2025b) at 4K and 8K resolutions, TWI-oriented TreeBench (Wang et al., 2025a), and multimodal reasoning benchmarks including MathVision (Wang et al., 2024) and LogicVista (Xiao et al., 2024).

**Implementation Details.** We evaluate our proposed RTWI using the SOTA TWI model Qwen3-VL (Bai et al., 2025)

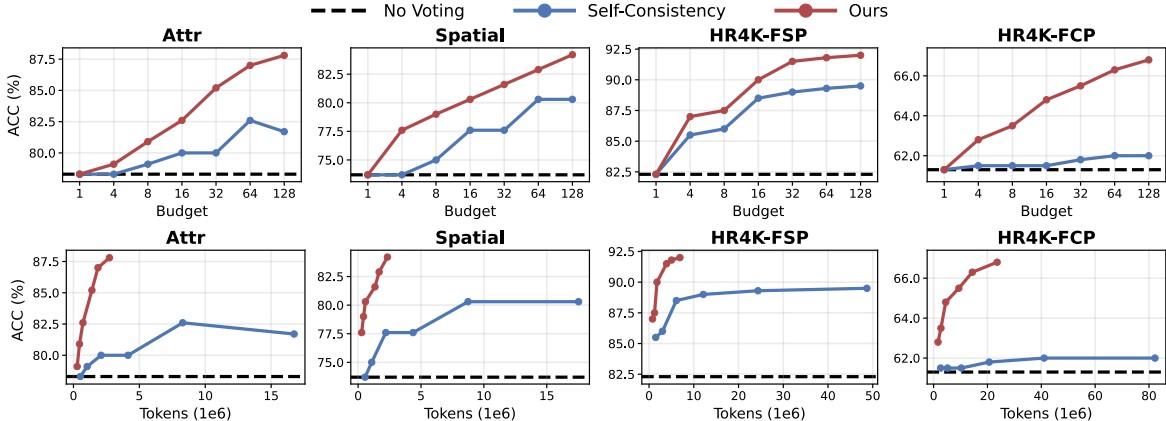

Figure 4. Test-time Scaling. The first row indicates the accuracy as the budget increases. The second row illustrates the relationship between accuracy and generated tokens under various test-time costs.

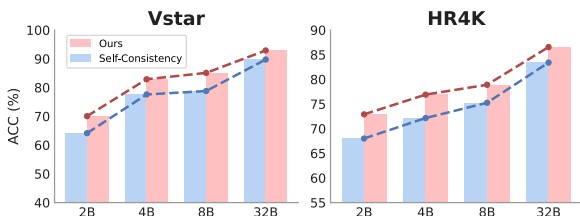

Figure 5. Scaling law across model sizes.

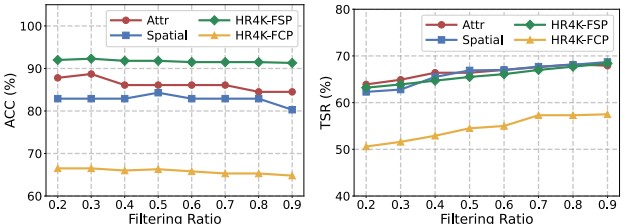

Figure 6. The parameter analysis of filtering ratio $\alpha$ in Eq 5.

and conduct comprehensive experiments on two variants, *i.e.*, Qwen3-VL-Thinking and Qwen3-VL-Instruct. Regarding hyperparameter settings, the filtering ratio $\alpha$ in Eq. 5 is set to $0.4$, while the temperature in Eq. 6 is fixed to $0.1$ for the Thinking variant and $1.0$ for the Instruct one.

### 4.2. Comparisons with State-Of-The-Arts

We compare RTWI with three SOTA TWI baselines, including GPT-4o (Achiam et al., 2023), Thyme (Zhang et al., 2025b), DeepEyes (Zheng et al., 2025), and six SOTA TTS scaling methods on different benchmarks under the online setting, including the vanilla Self-Consistency (SC) method (Wang et al., 2022), early stopping methods (*i.e.*, ASC (Aggarwal et al., 2023) and ESC (Li et al., 2024)), robust reasoning methods (*i.e.*, CISC (Taubenfeld et al., 2025), DeepConf (Fu et al., 2025), and Self-Cer. (Kang et al., 2025)). Specifically, in the online setting, following DeepConf (Fu et al., 2025), RTWI adopts an early-stopping strategy once the generated traces reach a consensus threshold $\beta = 0.9$, thus reducing the trace generation budget. Besides, for methods not specially designed for the online setting, we employ the same early-stopping strategy to enable their evaluation under online setting. For fair comparisons, we pre-generate 32 complete thinking traces for each question, and use them for evaluations among various baselines. For more results under offline setting, please refer

to Appendix C.3.

From the results in Table 1-3, one could have the following observations and conclusions: i) existing methods yield marginal performance gains over the vanilla self-consistency method, which could be attributed to the inability to identify and mitigate NT. In contrast, RTWI achieves robust filtering and voting against NT, thus significantly outperforming all the baselines across various model variants and benchmarks; ii) compared with existing early stopping and robust reasoning baselines, RTWI significantly avoids unnecessary token generation and achieves notable gains in computational efficiency, which might be attributed to our advanced reliability estimation of traces; iii) as shown in Table 2, RTWI achieves higher mean IOU (see more details in Appendix A.4) between the predicted and ground-truth visual cues of filtered traces, demonstrating that RTWI improves the final accuracy by excluding traces with unreliable visual cues.

### 4.3. Ablation and Analytic Study

In this section, we carry out a series of ablation and analytic experiments to investigate the effectiveness of RTWI. Unless otherwise stated, all the experiments are conducted under the online setting using the Qwen3-VL-8B-Thinking.

**Test-time Scaling.** To comprehensively investigate the effectiveness of RTWI, we conduct experiments using vari-

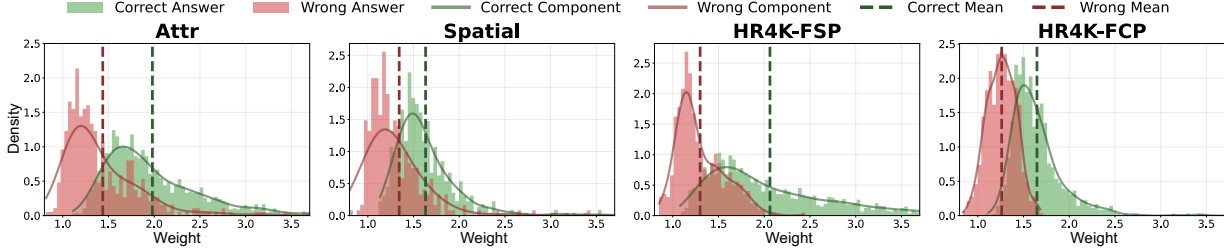

*Figure 7.* Analytical study of confidence distributions for correct and incorrect thinking traces.

*Table 4.* Ablation study of the designs in RTWI, where "w/o DF" denotes using only the consensus-based filtering, "w/o RV" denotes no usage of reliable voting.

| Variants | Vstar Bench | | HR-Bench 4K | |
|---|---|---|---|---|
| | ACC | TSR | ACC | TSR |
| SC | 78.8 | - | 75.3 | - |
| w/o DF | 80.8 | 35.6 | 77.1 | 38.4 |
| w/o RV | 81.9 | 61.2 | 77.9 | 56.2 |
| w/o $w_t$ | 82.3 | 58.3 | 77.9 | 50.2 |
| w/o $\Delta_t$ | 82.3 | 57.5 | 78.0 | 54.1 |
| Default | 83.4 | 61.2 | 78.6 | 56.2 |

*Table 5.* Analytic study on the ratio of high-entropy tokens in the tool-invocation and vanilla periods.

| Period | Vstar Bench | HR-Bench 4K |
|---|---|---|
| Tool Invocation | 68.6 | 66.0 |
| Vanilla | 31.4 | 34.0 |

*Table 6.* Analytic study about visual cue consistency.

| Method | Vstar Bench | | HR-Bench 4K | |
|---|---|---|---|---|
| | Consistency | ACC | Consistency | ACC |
| SC | 43.0 | 78.8 | 48.4 | 75.3 |
| CISC | 44.5 | 76.2 | 49.3 | 73.5 |
| DeepConf | 44.9 | 79.0 | 50.0 | 76.5 |
| Self-Cer. | 44.1 | 75.5 | 49.0 | 74.3 |
| Ours | **47.6** | **83.4** | **51.7** | **78.6** |

ous test-time budgets and model sizes. From the results in Fig. 4, one could have the following conclusions: i) RTWI yields significant performance gains over the vanilla self-consistency as the budget scales, demonstrating a favorable scaling behavior with respect to computational budget; ii) RTWI exhibits favorable efficiency-accuracy trade-offs as generated token scales. Moreover, we evaluate the proposed RTWI using Qwen3-VL Thinking models with 2B, 4B, 8B, and 32B parameters. As illustrated in Fig. 5, RTWI achieves consistent performance improvements as the model size scales, which aligns with the expected scaling law (Kaplan et al., 2020). In particular, the performance gains are more pronounced for models with limited reasoning capacity, *e.g.*, Qwen3-VL-2B-Thinking.

**Ablation studies.** To verify the importance of each design, we investigate the variants of the RTWI in Table 4, where one could have the following conclusions. On the one hand, both the proposed Dual-stage Filtering (DF) module and Reliable Voting (RV) module could select and prioritize reliable traces and thus boost the reasoning accuracy. On the other hand, the two-fold reliability principles could complement each other and thus facilitate accurate answer derivation and unreliable trace exclusion. As a result, RTWI could achieve better reasoning performance while saving more test-time compute. Besides, we carry out parameter analysis of the filtering ratio $\alpha$ in Eq. 5. As depicted in Fig. 6, RTWI demonstrates stable performance with low token consumption when $\alpha$ falls within the range of $[0.3, 0.5]$.

**Analytic Study on Reliability Estimation.** As pointed out

in Section 3.2, we estimate the reliability of visual cues by considering high-entropy tokens in textual CoT. To verify this claim, we conduct an analytic study on the distribution of Top-$k$ high-entropy tokens in the mining stage. As shown in Table 5, the ratio of high-entropy tokens in tool invocation is consistently higher than that in vanilla generation, indicating that RTWI focuses on tool-invocation tokens and thus effectively estimates the reliability of visual cues.

**Analytic Study on Dual-stage Filtering.** To demonstrate the effectiveness of the dual-stage filtering module, we conduct an analytical study on the visual cue consistency among the filtered traces $T_{\text{rel}}$, see more details in Appendix A.5. As shown in Table 6, RTWI achieves the highest consistency compared with the most competitive methods in Table 1-3, which indicates that RTWI improves reasoning accuracy by prioritizing its focus on similar task-relevant regions.

**Analytic Study on Reliable Voting.** To verify the effectiveness of the reliable voting module, we visualize the confidence distribution of traces with correct or incorrect answers. From the results in Fig. 7, traces with correct answers generally exhibit higher confidence than those with incorrect answers, indicating that confidence serves as an effective indicator for identifying NT.

# 5. Conclusion

In this paper, we study a new problem in TWI, i.e., Noisy Thinking, which refers to incorrect cue mining and answer reasoning stages. To solve this problem, the proposed RTWI employs a unified text-centric reliability estimation mechanism to identify NT and accordingly mitigate the negative impact of NT by reliability-aware filtering and voting. Extensive experiments on real-world benchmarks under offline and online settings verify the effectiveness of RTWI against NT. In the future, we plan to explore the reliable TTS paradigm in a broader range of agent applications, advancing agentic systems toward reliable reasoning.

# Impact Statement

This paper presents work whose goal is to advance the thinking with images paradigm towards reliable reasoning. There are many potential societal consequences of our work, none of which we feel must be specifically highlighted here.

# Author Contributions

All authors contributed significantly to this work. Xi Peng and Mouxing Yang conceived the study, designed the RTWI algorithm, refined the manuscript, and supervised the project. Haobin Li co-designed and implemented the RTWI algorithm, conducted the baseline evaluations, and drafted the manuscript. Yutong Yang, Yijie Lin, and Xiang Dai analyzed the experimental results and contributed to the formulation of the manuscript. All authors reviewed and approved the final version.

# Acknowledgments

This work was supported in part by NSFC under Grant 624B2099 and U24B20174; in part by the Fundamental Research Funds for the Central Universities under Grant CJ202303; in part by Sichuan Science and Technology Planning Project under Grant 24NSFTD0130; in part by Fundamental and Interdisciplinary Disciplines Breakthrough Plan of the Ministry of Education of China under Grant JYB2025XDXM610; and in part by Foundation Enhancement Program Project (Technology Field Fund) under Grant 2025-JCJQ-JJ-0686.

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

# A. More Implementation Details

In this section, we provide more implementation details about the reliability estimation, online filtering, mIoU metric, and generation hyperparameter.

## A.1. More Details about Reliability Estimation for Single-Stage Traces

Without requiring external supervision (Kou et al., 2025b;a; Feng et al., 2025a), we estimate the reliability of the thinking traces using model-intrinsic signals, *i.e.*, token entropy. However, as discussed in (Zheng et al., 2025; Zhang et al., 2025b), MLLM would adaptively decide whether to adopt external tools for visual cue extraction based on the difficulty of the question. Accordingly, not all generated traces involve tool usage, making it infeasible to derive confidence weights for all traces using Eq. 6. To remedy this, for a given single-stage trace $t = \{t^r\}$, we estimate its confidence weight as follows,

$$C_t = exp\left(\left(\eta^r - \eta^m\right) / \left(|w_t| \cdot \tau\right)\right), \tag{9}$$

where $w_t = 2w(t^r)$. Such a formulation not only assigns limited confidence weight to unreliable single-stage traces, but also remains consistent with the weighting mechanism in Eq. 6.

## A.2. More Details about Dual-stage Filtering under Online Setting

In the online setting, TWI generates multiple turns of interleaved image-text CoT and then the final turn with textual CoT. However, during online generation, the stage of the current turn is unknown. As a result, it is infeasible to employ the stage-specific thresholds to select reliable traces. To remedy this, at each turn $i$, we dynamically estimate the reliability of the currently generated textual CoT $\mathbf{x}_i^{\text{txt}}$, *i.e.*,

$$w(\mathbf{x}_i^{\text{txt}}) = -\frac{1}{k} \sum_{j \in \mathcal{K}\left(\mathcal{H}(\mathbf{x}_i^{\text{txt}})\right)} h_j, \tag{10}$$

where $\mathcal{H}(\mathbf{x}_i^{\text{txt}})$ denotes the set of token entropies at the $i$-th turn. During online generation, we perform a preliminary filtering using the loose threshold $\eta^m$, which is motivated by the empirical observation that $\eta^m < \eta^r$ as shown in Fig. 3. For a given thinking trace $t$ with multi-turn textual CoT $\{\mathbf{x}_i^{\text{txt}}\}_{i=1}^N$, the trace is preserved during the online generation only if

$$w(\mathbf{x}_i^{\text{txt}}) \geq \eta^m, \quad \forall i = 1, \ldots, N. \tag{11}$$

After online filtering, the remaining traces are further subjected to the more stringent dual-stage filtering described in Eq. 4 to determine the final set of reliable traces.

## A.3. More Details about Reliability Estimation in Eq. 3

The number $k$ for reliability estimation in Eq. 3 is adaptively determined as

$$\arg\max_k \sum_{t \in T_{\text{sel}}} \left(w(t^r) - w(t^m)\right), \tag{12}$$

which maximizes the reliability leap to facilitate the identification of reliable traces. Besides, following (Holtzman et al., 2019), token entropy is computed by employing the normalized distribution over the Top-10 token probabilities.

## A.4. More Details about the mIoU Metric

Unlike benchmarks (Wang et al., 2025b; Wu & Xie, 2024) that could only quantify multimodal reasoning capabilities (Lin et al., 2026; Feng et al., 2025b) through final outcomes. The TWI-oriented TreeBench provides reference localizations of key visual cues, *i.e.*, bounding boxes, which could be employed to evaluate the effectiveness of the proposed robust filtering mechanism. Following (Wang et al., 2025a), we employ the mean Intersection-Over-Union (mIoU) metric to measure the alignment between predicted and ground-truth visual cues.

Specifically, for a given multimodal question with ground-truth bounding boxes $\{b_k\}_{k=1}^M$, each filtered trace $t \in T_{\text{rel}}$ generates a set of predicted bounding boxes $\{\hat{b}_i^t\}_{i=1}^{N_t}$. Accordingly, we define the trace-level IoU as

$$\text{IoU}(t) = \frac{1}{N_t} \sum_{i=1}^{N_t} \text{IoU}\left(\{b_k\}_{k=1}^M, \hat{b}_i^t\right), \tag{13}$$

where $\text{IoU}\left(\{b_k\}_{k=1}^M, \hat{b}_i^t\right) = \max_k \text{IoU}\left(b_k, \hat{b}_i^t\right)$. After that, the overall mIoU score over the filtered trace set $T_{\text{rel}}$ is derived as

$$\text{mIoU}(T_{\text{rel}}) = \frac{1}{|T_{\text{rel}}|} \sum_{t \in T_{\text{rel}}} \text{IoU}(t). \tag{14}$$

The formulation of mIoU reflects the intuition that each predicted bounding box should correspond to at least one ground-truth visual cue.

### A.5. More Details about Visual Cue Consistency

In the manuscript, we conduct an analytical study on the visual cue consistency among the filtered traces. Specifically, given the filtered trace set $T_{\text{rel}}$, we define visual cue consistency as the average IoU between pairwise visual cues among the filtered traces, *i.e.*,

$$\text{Cons}(T_{\text{rel}}) = \frac{1}{\displaystyle\sum_{\substack{t,t' \in T_{\text{rel}} \\ t \neq t'}} N_t N_{t'}} \sum_{\substack{t,t' \in T_{\text{rel}} \\ t \neq t'}} \sum_{i=1}^{N_t} \sum_{j=1}^{N_{t'}} \text{IoU}\left(\hat{b}_i^t, \hat{b}_j^{t'}\right). \tag{15}$$

where $\{\hat{b}_i^t\}_{i=1}^{N_t}$ denote the predicted bounding boxes produced by trace $t$.

### A.6. Generation Hyperparameters

We list the per-model decoding hyperparameters used across all experiments. For each model, we fix the temperature, top-$p$, top-$k$, and the maximum generation length.

*Table 7.* Generation hyperparameters used in our experiments. Different models use different decoding settings.

| Model | Temperature | Top-$p$ | Top-$k$ | Max seq len |
|---|---|---|---|---|
| Qwen3-VL-Thinking | 0.6 | 0.95 | 20 | 51200 |
| Qwen3-VL-Instruct | 1.0 | 1.0 | 0 | 51200 |
| DeepEyes | 1.0 | 1.0 | 0 | 51200 |

# B. Pseudocode

In this section, we provide pseudocode of the proposed RTWI under the online and offline settings.

---

**Algorithm 1** Reliable Online TWI

---

1: **Inputs:** Multimodal question $\{Q, I\}$, budget $B$, filtering ratio $\alpha$, threshold $\beta$, $i \leftarrow 0$.
2: **Offline Warm-up:**
3: Generate warm-up traces $T_{\text{sel}}$ for $\{Q, I\}$ , initialize $(\eta^m, \eta^r)$ using Eq. 5 with $\alpha$, initialize $T_{\text{rel}}$ using Eq. 4.
4: Compute voting values $V(a) = \sum_{t \in T_{\text{rel}}} C_t \cdot \mathbb{I}[A_t = a]$ for all answer $a$ and majority answer $\hat{a} = \arg\max_a V(a)$.
5: **Online Generation:**
6: **while** $V(\hat{a}) / \sum_a V(a) < \beta$ **and** $|T_{\text{sel}}| + i < B$ **do**
7:     **while** generating trace $t_i$ **do**
8:         Generate token $j$ and update reliability $w(t_i^s)$ for $s \in \{m, r\}$.
9:         **If** $\exists s,\ w(t_i^s) < \eta^s$: stop generating $t_i$; **else**: add $j$ to $t_i$.
10:     **end while**
11:     Update $i \leftarrow i + 1$. **If** $t_i$ is completed: update $T_{\text{rel}} \leftarrow T_{\text{rel}} \cup \{t_i\}$, $V(a)$ and $\hat{a}$.
12: **end while**
13: **return** Final answer $\hat{A}$.

---

---

**Algorithm 2** Reliable Offline TWI

---

1: **Inputs:** Multimodal question $\{Q, I\}$, number of traces $B$, filtering ratio $\alpha$.
2: Initialize trace set $T \leftarrow \emptyset$.
3: **for** $i = 1$ **to** $B$ **do**
4:     Generate a complete thinking trace $t_i$ for $\{Q, I\}$.
5:     $T \leftarrow T \cup \{t_i\}$.
6: **end for**
7: Calculate thresholds $(\eta^m, \eta^r)$ using Eq. 5 with $T$ and $\alpha$, and then construct reliable trace set $T_{\text{rel}}$ using Eq. 4.
8: Determine the final answer $\hat{A}$ using Eq. 6 with $T_{\text{rel}}$.
9: **return** Final answer $\hat{A}$.

---

# C. More Experimental Results

In this section, we present more experimental results of the proposed RTWI. Unless otherwise specified, all experiments are conducted using the Qwen3-VL-8B-Thinking model.

## C.1. More Experiments on Various TWI Models

In the manuscript, we have carried out experiments using Qwen3-VL-8B-Thinking and Qwen3-VL-8B-Instruct. To further verify the effectiveness of RTWI, we provide more experimental results under online setting using the widely-used TWI model DeepEyes-7B (Zheng et al., 2025). From the results in Table 8, RTWI outperforms all baselines across various datasets in terms of reasoning accuracy and efficiency.

*Table 8.* Results on real-world high-resolution benchmarks under online setting using DeepEyes-7B (Zheng et al., 2025) as backbone. Base* indicates our reproduced performance of DeepEyes-7B.

| | Vstar Bench | | | | HR-Bench 4K | | | | HR-Bench 8K | | | |
| | Attr | | Spatial | | FSP | | FCP | | FSP | | FCP | |
| Method | ACC | TSR | ACC | TSR | ACC | TSR | ACC | TSR | ACC | TSR | ACC | TSR |
|---|---|---|---|---|---|---|---|---|---|---|---|---|
| GPT-4o | 72.2 | - | 60.5 | - | 66.8 | - | 63.3 | - | 60.8 | - | 58.5 | - |
| Thyme | 83.5 | - | 80.3 | - | 91.0 | - | 63.0 | - | 86.5 | - | 57.5 | - |
| *DeepEyes-7B* | | | | | | | | | | | | |
| Base* | 86.1 | - | 80.3 | - | 87.8 | - | 56.5 | - | 81.3 | - | 55.3 | - |
| SC | 86.1 | - | 80.3 | - | 89.5 | - | 58.0 | - | **86.0** | - | 57.3 | - |
| ASC | 85.2 | 60.0 | 80.3 | 58.1 | 89.5 | 58.9 | 58.0 | 48.9 | **86.0** | 58.1 | 57.3 | 47.4 |
| ESC | 86.1 | 55.0 | 80.3 | 48.2 | 89.5 | 55.8 | 58.0 | 31.3 | **86.0** | 47.2 | 57.3 | 24.7 |
| CISC | **87.0** | 55.4 | 80.3 | 51.8 | 89.8 | 56.4 | 56.8 | 37.1 | 85.8 | 50.6 | 56.8 | 34.0 |
| DeepConf | 86.1 | 58.2 | 79.0 | 56.8 | 89.5 | 53.9 | 58.0 | 48.0 | **86.0** | 53.7 | 58.3 | 43.9 |
| Self-Cer. | 85.2 | 62.5 | 80.3 | 56.8 | 89.5 | 61.9 | 58.0 | 40.2 | 85.3 | 57.5 | 56.5 | 36.5 |
| Ours | **87.0** | **67.5** | **81.6** | **66.9** | **90.0** | **64.5** | **60.0** | **53.4** | **86.0** | **63.5** | **59.0** | **50.1** |

## C.2. More Experiments on VisualProbe Benchmark

In the manuscript, we have conducted experiments on the TWI-oriented TreeBench. Here, we carry out additional experiments on the VisualProbe (Lai et al., 2025) benchmark. As shown in Table 9, RTWI achieves higher reasoning accuracy and efficiency than most baselines across different difficulty levels.

*Table 9.* Results on VisualProbe Benchmark.

| | Easy | | Medium | | Hard | |
| Method | ACC | TSR | ACC | TSR | ACC | TSR |
|---|---|---|---|---|---|---|
| GPT-4o | 47.5 | - | 15.4 | - | 11.2 | - |
| DeepEyes | 60.1 | - | 29.8 | - | 35.1 | - |
| *Qwen3-VL Thinking* | | | | | | |
| Base | 47.6 | - | 25.8 | - | 28.3 | - |
| SC | 56.8 | - | 29.6 | - | 30.2 | - |
| ASC | 56.8 | 37.2 | 29.6 | 32.6 | 30.2 | 29.7 |
| ESC | 56.8 | 11.4 | 29.6 | 10.9 | 30.2 | 8.6 |
| CISC | 52.5 | 22.7 | 27.7 | 20.4 | 27.4 | 15.0 |
| DeepConf | 56.8 | 42.9 | 29.6 | 41.0 | 29.2 | 40.3 |
| Self-Cer. | 54.0 | 23.9 | 27.3 | 21.4 | 27.4 | 14.5 |
| Ours | **60.4** | **44.3** | **34.1** | 41.9 | 30.2 | **44.6** |
| *Qwen3-VL Instruct* | | | | | | |
| Base | 56.0 | - | 35.4 | - | 34.9 | - |
| SC | 64.0 | - | 40.2 | - | 42.5 | - |
| ASC | 64.0 | **20.7** | 40.2 | **21.8** | 42.5 | **22.5** |
| ESC | 64.0 | 4.7 | 40.2 | 8.8 | 42.5 | 6.7 |
| CISC | 59.7 | 12.8 | 38.6 | 14.7 | 39.6 | 9.2 |
| DeepConf | 61.9 | 14.6 | **40.6** | 16.7 | **43.4** | 14.7 |
| Self-Cer. | 59.0 | 13.9 | 39.4 | 14.5 | 37.7 | 12.0 |
| Ours | **65.4** | 16.2 | **40.6** | 18.5 | 42.5 | 16.2 |

## C.3. Offline Evaluations

In the manuscript, we have conducted experiments under online setting. Here, we further provide more implementation details and experimental results on various benchmarks under offline setting. Specifically, Alg. 2 provides the details of the algorithm under offline setting. From the results in Table 10-12, one could have the following conclusions: i) RTWI consistently outperforms all baselines across different benchmarks under offline setting, demonstrating its robustness against NT; ii) comparing performances between the online and offline settings, the online implementation achieves similar results to its offline counterpart, indicating that the proposed warm-up strategy could effectively estimate the reliability thresholds and thus facilitate dual-stage filtering and voting.

*Table 10.* Results on real-world high-resolution benchmarks under offline setting.

| Method | Vstar Bench | | HR-Bench 4K | | HR-Bench 8K | |
|---|---|---|---|---|---|---|
| | Attr | Spatial | FSP | FCP | FSP | FCP |
| *Qwen3-VL Thinking* | | | | | | |
| Base | 78.3 | 73.7 | 83.0 | 57.3 | 78.3 | 56.0 |
| SC | 80.0 | 77.6 | 89.5 | 61.0 | 86.5 | 61.0 |
| CISC | 84.3 | 69.7 | 87.8 | 59.8 | 88.5 | 56.0 |
| DeepConf | 81.7 | 76.3 | 89.5 | 62.5 | 87.0 | 61.8 |
| Self-Cer. | 82.6 | 68.4 | 89.5 | 61.0 | 88.0 | 59.0 |
| Ours | **86.1** | **82.9** | **91.5** | **65.5** | **89.0** | **63.0** |
| *Qwen3-VL Instruct* | | | | | | |
| Base | 90.4 | 86.8 | 93.8 | 72.0 | 90.8 | 71.3 |
| SC | 92.2 | 89.5 | 95.3 | 77.0 | 93.0 | 74.8 |
| CISC | 93.9 | 89.5 | 95.8 | 77.5 | 93.0 | 75.8 |
| DeepConf | 93.9 | **90.8** | 96.0 | 76.5 | 93.0 | 75.8 |
| Self-Cer. | 93.9 | **90.8** | 96.0 | 76.5 | 92.5 | 75.5 |
| Ours | **94.8** | **90.8** | **96.5** | **78.8** | **95.0** | **76.0** |

*Table 11.* Results on TWI-oriented TreeBench under offline setting.

| Method | Reasoning | | Perception | |
|---|---|---|---|---|
| | ACC | mIoU | ACC | mIoU |
| *Qwen3-VL Thinking* | | | | |
| Base | 32.8 | - | 63.1 | - |
| SC | 30.9 | 37.8 | 66.4 | 32.5 |
| CISC | 25.8 | 37.8 | 66.4 | 32.5 |
| DeepConf | 30.9 | 39.0 | 64.4 | 32.9 |
| Self-Cer. | 30.9 | 37.8 | 65.8 | 32.5 |
| Ours | **37.1** | **41.8** | **68.5** | **36.0** |
| *Qwen3-VL Instruct* | | | | |
| Base | 34.4 | - | 64.4 | - |
| SC | 34.8 | 43.4 | 68.4 | 38.6 |
| CISC | 34.8 | 43.4 | 68.4 | 38.6 |
| DeepConf | 35.6 | 43.6 | 69.8 | 38.9 |
| Self-Cer. | 35.6 | 43.4 | 69.8 | 38.6 |
| Ours | **35.9** | **46.8** | **71.1** | **42.0** |

*Table 12.* Results on reasoning benchmarks under offline setting.

| Method | MathVision | LogicVista |
|---|---|---|
| *Qwen3-VL Thinking* | | |
| Base | 18.7 | 44.3 |
| SC | 19.8 | 46.4 |
| CISC | 20.1 | 50.2 |
| DeepConf | 19.7 | 50.8 |
| Self-Cer. | 18.0 | 49.1 |
| Ours | **23.4** | **59.1** |
| *Qwen3-VL Instruct* | | |
| Base | 21.4 | 47.9 |
| SC | 24.4 | 54.8 |
| CISC | 22.7 | 55.3 |
| DeepConf | 22.7 | 55.4 |
| Self-Cer. | 24.0 | 55.4 |
| Ours | **25.5** | **60.3** |

## C.4. Analytic Study on Tool Usage

As mentioned in Section A.1, not all generated traces involve tool usage. Here, we conduct an additional analytical study to further examine tool usage with Qwen3-VL-8B-Thinking and Qwen3-VL-8B-Instruct as the TWI model. As illustrated in Fig. 8, the Instruct variant exhibits stronger instruction-following ability than the Thinking variant, leading to more frequent tool usage with a higher number of invocations. Notably, such a phenomenon indicates that the proposed RTWI could improve performance for both textual CoT and interleaved CoT with tool invocations.

## C.5. Analytic Study on CoT Length

As pointed out in Section 3.2, the lengths of textual CoT in the mining and reasoning stages are not necessarily consistent. Here, we conduct additional analytical experiments to examine stage-wise CoT lengths. As shown in Fig. 9, one could have

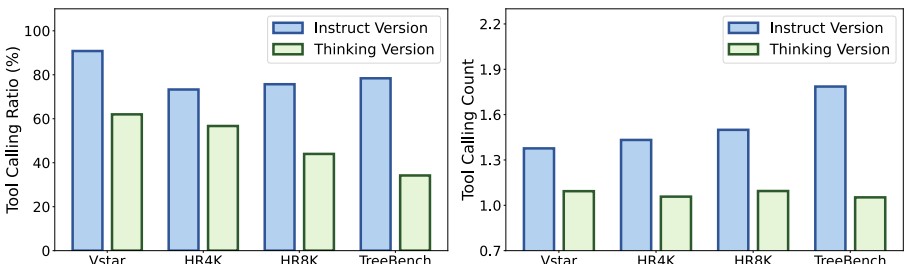

*Figure 8.* Analytic study on tool usage.

the following observations: i) the same TWI model generates textual CoTs of varying lengths across different datasets, *e.g.*, on more challenging benchmarks such as TreeBench, the model tends to generate more tokens; ii) the CoT length in the mining stage is generally longer than that in the reasoning stage, suggesting that TWI models allocate substantial computation to acquiring visual cues for facilitating the subsequent reasoning process; iii) the Thinking variant generates longer textual CoTs than the Instruct variant. Based on these observations, the proposed Top-$k$ selection strategy with a self-adaptively determined $k$ enables more stable reliability estimation across different datasets and TWI models.

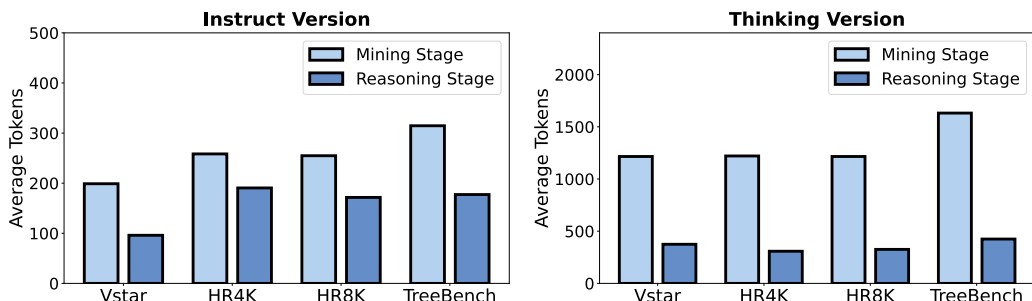

*Figure 9.* Analytic study on CoT length.

## C.6. Analytic Study on Saving Tokens

Here, we visualize token reduction patterns across various benchmarks in Fig. 10, illustrating how RTWI achieves substantial computational savings while preserving competitive accuracy.

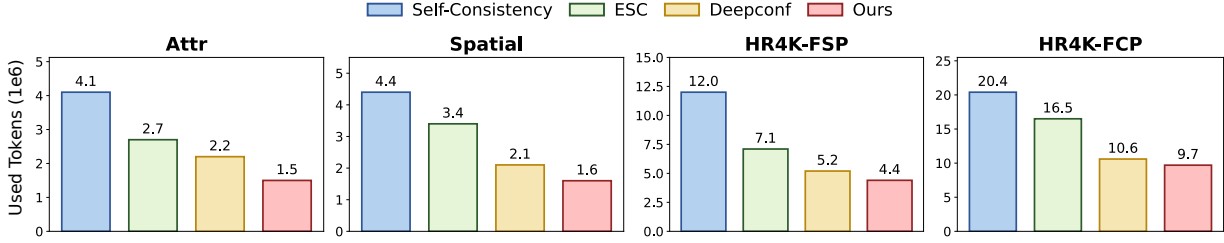

*Figure 10.* Generated tokens comparison across different benchmarks.

# D. More Ablation Studies

In this section, we present more ablation studies about the parameter analysis.

## D.1. Ablation Study on Temperature

We carry out an additional ablation study on the temperature $\tau$ in Eq. 6. From the results in Fig. 11, we observe that RTWI exhibits stable performance and token saving ratios when $\tau$ falls within the range of $[0.05, 0.1]$.

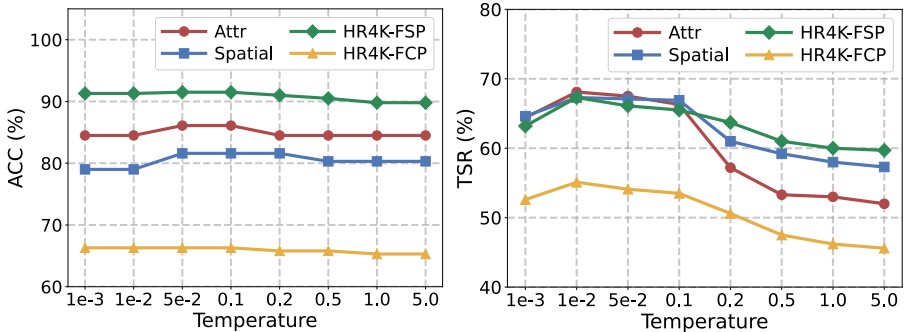

*Figure 11.* Ablation study on temperature $\tau$ in Eq. 6.

## D.2. Ablation on Warm-up Traces

We conduct an additional ablation study on the number of warm-up traces $|T_{\text{sel}}|$. As shown in Table 13, we observe that increasing $|T_{\text{sel}}|$ generally makes reasoning accuracy under the online setting closer to that under the offline setting, while reducing token saving ratios as RTWI could only employ early stopping on the remaining $B - |T_{\text{sel}}|$ traces.

*Table 13.* **Ablation Study on the Number of Warm-up Traces $|T_{\text{sel}}|$.**

| | | Vstar Bench | | | | HR-Bench 4K | | | |
| | | Attr | | Spatial | | FSP | | FCP | |
| $|T_{\text{sel}}|$ | Method | ACC | TSR | ACC | TSR | ACC | TSR | ACC | TSR |
|---|---|---|---|---|---|---|---|---|---|
| 4.0 | DeepConf | 80.0 | 64.0 | 77.6 | 67.5 | 89.0 | 63.6 | 60.3 | 64.9 |
| | Ours | 86.1 | 74.6 | 80.3 | 72.6 | 90.0 | 75.3 | 64.0 | 66.3 |
| 8.0 | DeepConf | 81.7 | 47.6 | 76.3 | 52.8 | 90.5 | 57.3 | 62.5 | 48.9 |
| | Ours | 85.2 | 61.1 | 81.6 | 61.2 | 91.3 | 61.9 | 65.8 | 50.5 |
| 12.0 | DeepConf | 81.7 | 44.2 | 76.3 | 46.4 | 90.5 | 48.6 | 62.0 | 30.5 |
| | Ours | 86.1 | 48.2 | 82.9 | 51.4 | 91.5 | 53.8 | 66.0 | 31.7 |
| 16.0 | DeepConf | 81.7 | 36.7 | 77.6 | 37.5 | 91.5 | 27.9 | 62.5 | 26.2 |
| | Ours | 86.1 | 38.5 | 81.6 | 41.1 | 91.5 | 30.4 | 66.0 | 30.1 |

# E. Evaluation Prompts

To ensure reproducibility and facilitate future research, we provide the prompts used to evaluate RTWI across all benchmarks. Note that the prompts are adapted from the official prompts of Qwen3-VL (Bai et al., 2025).

---

**Prompt for Multiple-Choice Benchmarks**

- Your role is that of a research assistant specializing in visual information. Answer questions about images by looking at them closely and providing detailed analysis. Please follow this structured thinking process and show your work.

- Start an iterative loop for each question:

    - **First, look closely:** Begin with a detailed description of the image, paying attention to the user's question. List what you can tell just by looking, and what you'll need to look up.

    - **Next, find information:** Use a tool to research the things you need to find out.

    - **Then, review the findings:** Carefully analyze what the tool tells you and decide on your next action.

- Continue this loop until your research is complete.

- To finish, put your final answer within \boxed{}. Answer with the option's letter from the given choices directly, e.g. \boxed{A}, \boxed{B}, \boxed{C}, \boxed{D}, etc.

---

**Prompt for Open-Ended Benchmarks**

- Your role is that of a research assistant specializing in visual information. Answer questions about images by looking at them closely and providing detailed analysis. Please follow this structured thinking process and show your work.

- Start an iterative loop for each question:

    - **First, look closely:** Begin with a detailed description of the image, paying attention to the user's question. List what you can tell just by looking, and what you'll need to look up.

    - **Next, find information:** Use a tool to research the things you need to find out.

    - **Then, review the findings:** Carefully analyze what the tool tells you and decide on your next action.

- Continue this loop until your research is complete.

- To finish, You MUST PUT your FINAL ANSWER within \boxed{}, and make sure it contains only the answer itself without extra words or symbols.

---

# F. Case Study

In this section, we provide some examples of successful identification of NT, as well as cases of failure. As shown in Fig. 12–19, RTWI could effectively identify reliable and noisy thinking processes by estimating reliability for both the visual cues in the mining stage and the textual CoT in the reasoning stage.

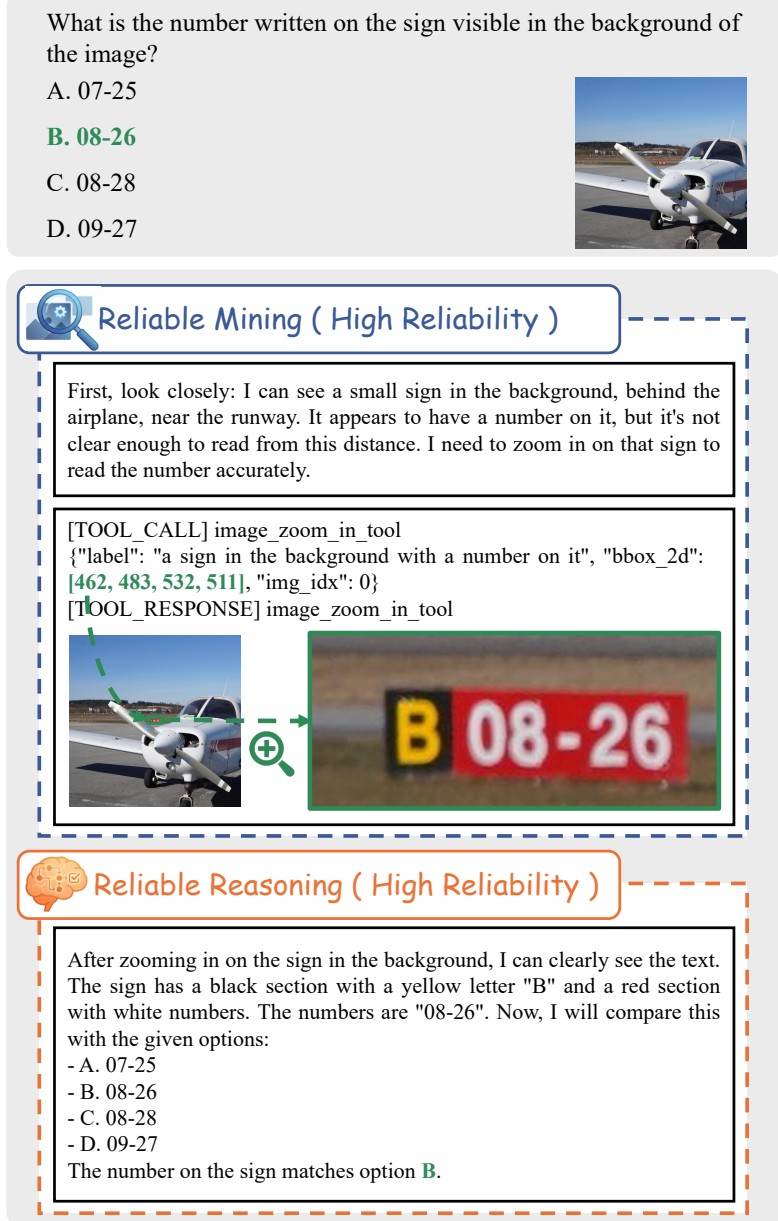

*Figure 12.* Example of reliable trace: visual perception and OCR. The TWI model correctly perceives the sign and performs OCR to recognize the numbers displayed on it.

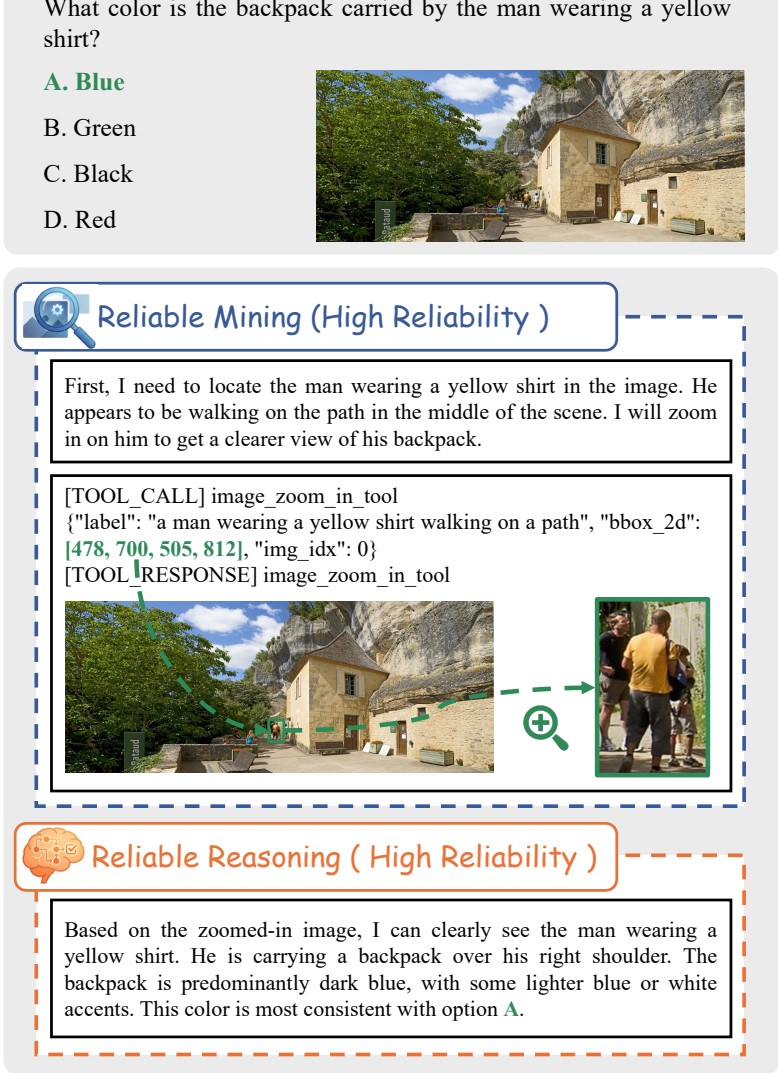

*Figure 13.* Example of reliable trace: visual perception and color identification. The TWI model identifies the man wearing a yellow shirt and correctly recognizes the color of his backpack.

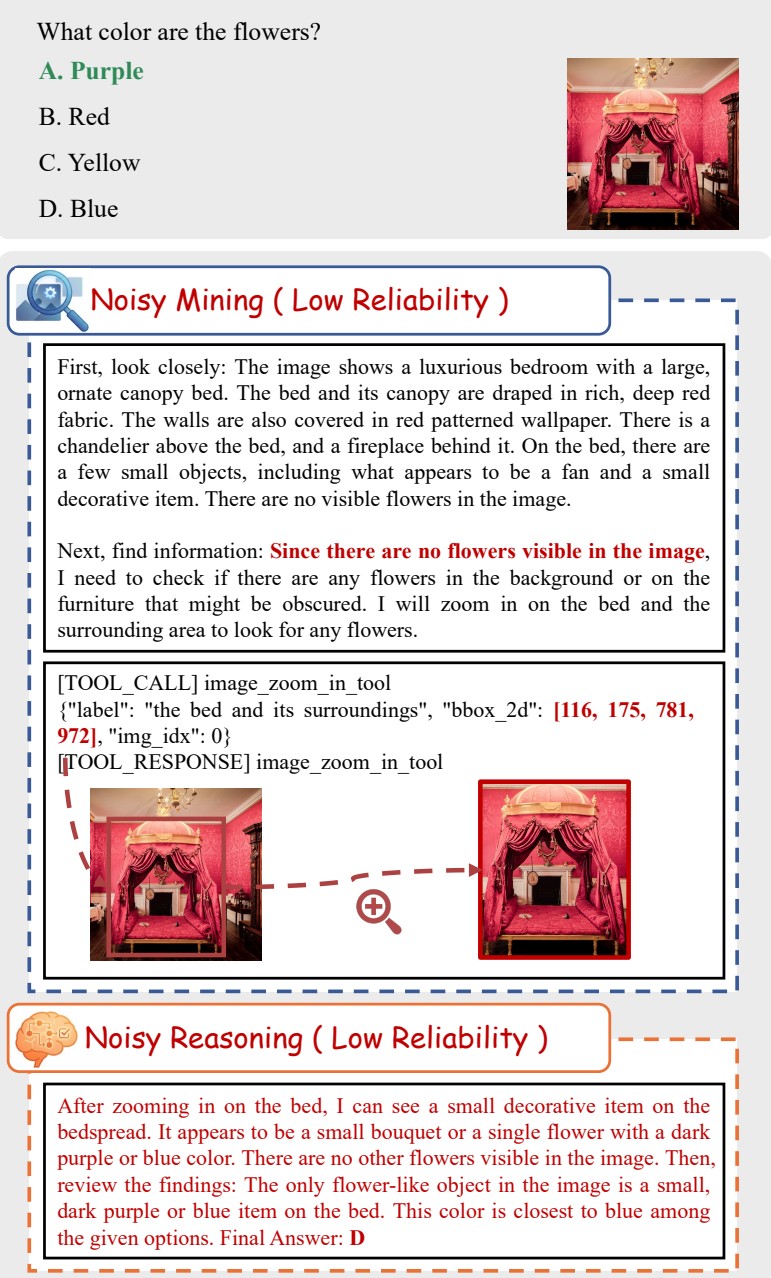

*Figure 14.* Example of noisy mining: failed visual perception. The TWI model locates the coarse-grained region but fails to derive the fine-grained flowers required for correct reasoning.

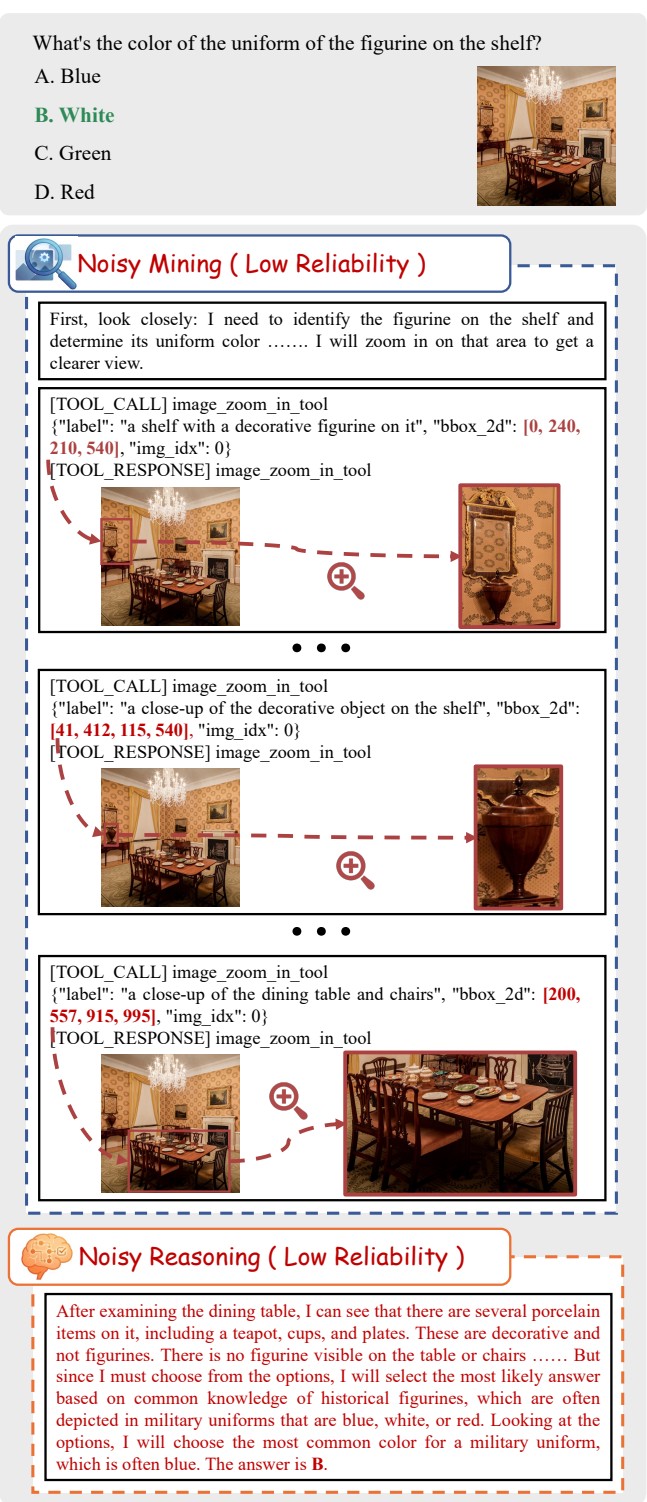

*Figure 15.* Example of noisy mining: failed visual search. The TWI model repeatedly searches the image but fails to locate the figurine in the thinking stage.

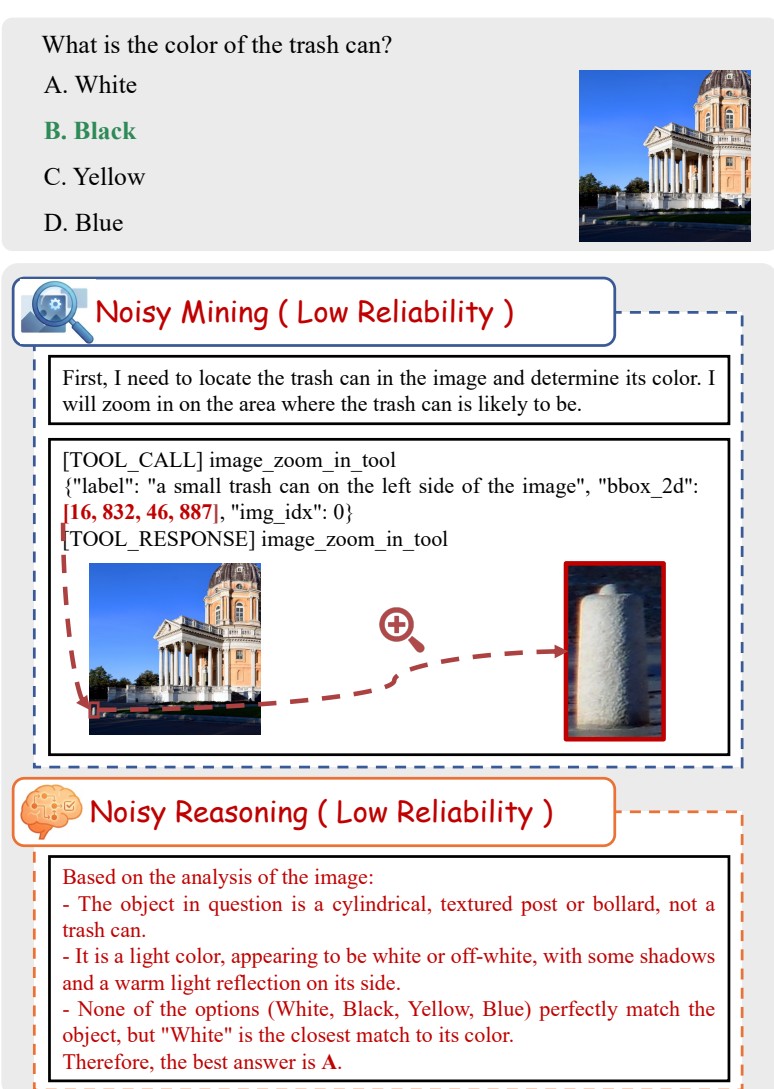

*Figure 16.* Example of noisy mining: false positive visual perception. The TWI model mistakenly identifies a stone pillar as a trash can in the mining stage, then recognizes the visual cue is incorrect yet still derives the wrong answer during the reasoning stage.

What is the position of the black car relative to the tram?

A. To the left of the tram

**B. Behind the tram**

C. In front of the tram

D. To the right of the tram

### 🔍 Reliable Mining ( High Reliability )

First, I need to locate both the black car and the tram in the image to determine their relative positions. I will start by zooming in on the tram to get a clear view of its location.

[TOOL_CALL] image_zoom_in_tool
{"label": "the tram in the center of the street", "bbox_2d": **[488, 547, 637, 863]**, "img_idx": 0}
[TOOL_RESPONSE] image_zoom_in_tool

Now I have a clear view of the tram. It is a double-decker tram with the number 23 on its roof and front. I can see it is moving down the street, surrounded by crowds and market stalls. Now I need to locate the black car in the image to determine its position relative to the tram. I will now scan the image to find the black car, focusing on the upper part of the image where vehicles are more likely to be.

[TOOL_CALL] image_zoom_in_tool
{"label": "a black car parked on the upper left side of the street", "bbox_2d": **[287, 87, 392, 143]**, "img_idx": 0}
[TOOL_RESPONSE] image_zoom_in_tool

### 🧠 Noisy Reasoning ( Low Reliability )

Based on the analysis of the image:
1. The tram is located in the center of the street, moving down the road.
2. The black car is located in the upper left portion of the image, parked on the side of the road.
3. **From the perspective of the viewer, the black car is positioned to the left of the tram.**
Therefore, the position of the black car relative to the tram is to its left.
The correct option is: **A**. To the left of the tram

*Figure 17.* Example of noisy reasoning: limited understanding of perspective projection. Although the TWI model correctly locates both the tram and the black car, it fails to account for the visual perspective and spatial mapping between them in the reasoning stage.

Is the baby carriage on the left or right side of the cone?

**A. The baby carriage is on the right side of the cone.**

B. The baby carriage is on the left side of the cone.

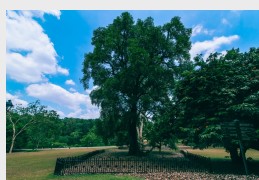

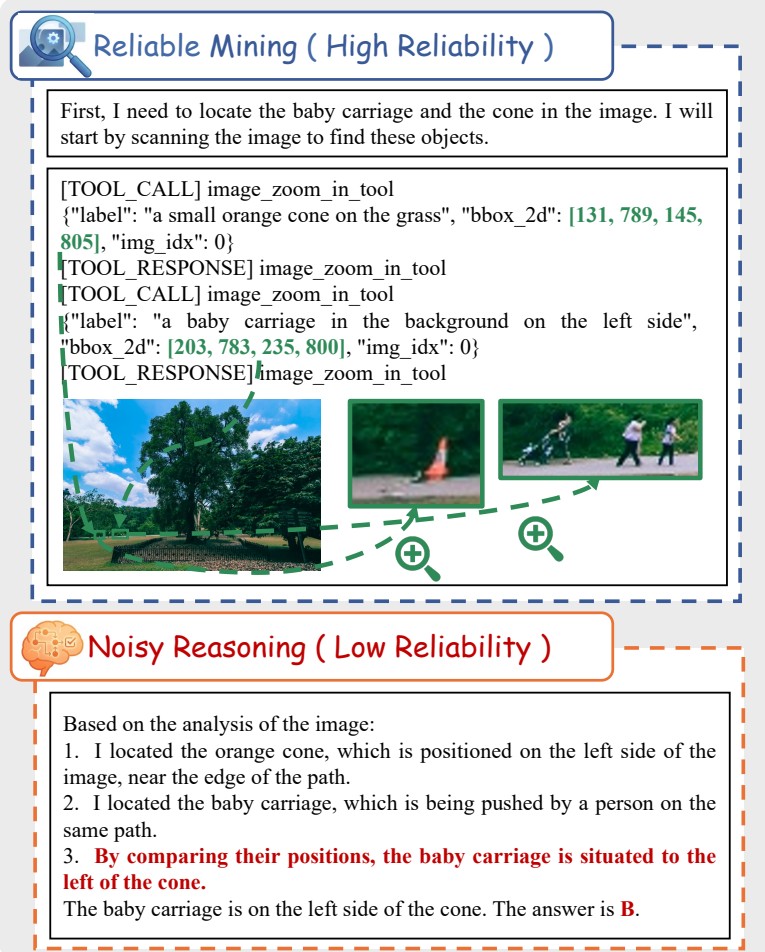

*Figure 18.* Example of noisy reasoning: limited spatial understanding. Although the TWI model successfully locates both the baby carriage and the cone, it misinterprets the spatial relationship in the reasoning stage.

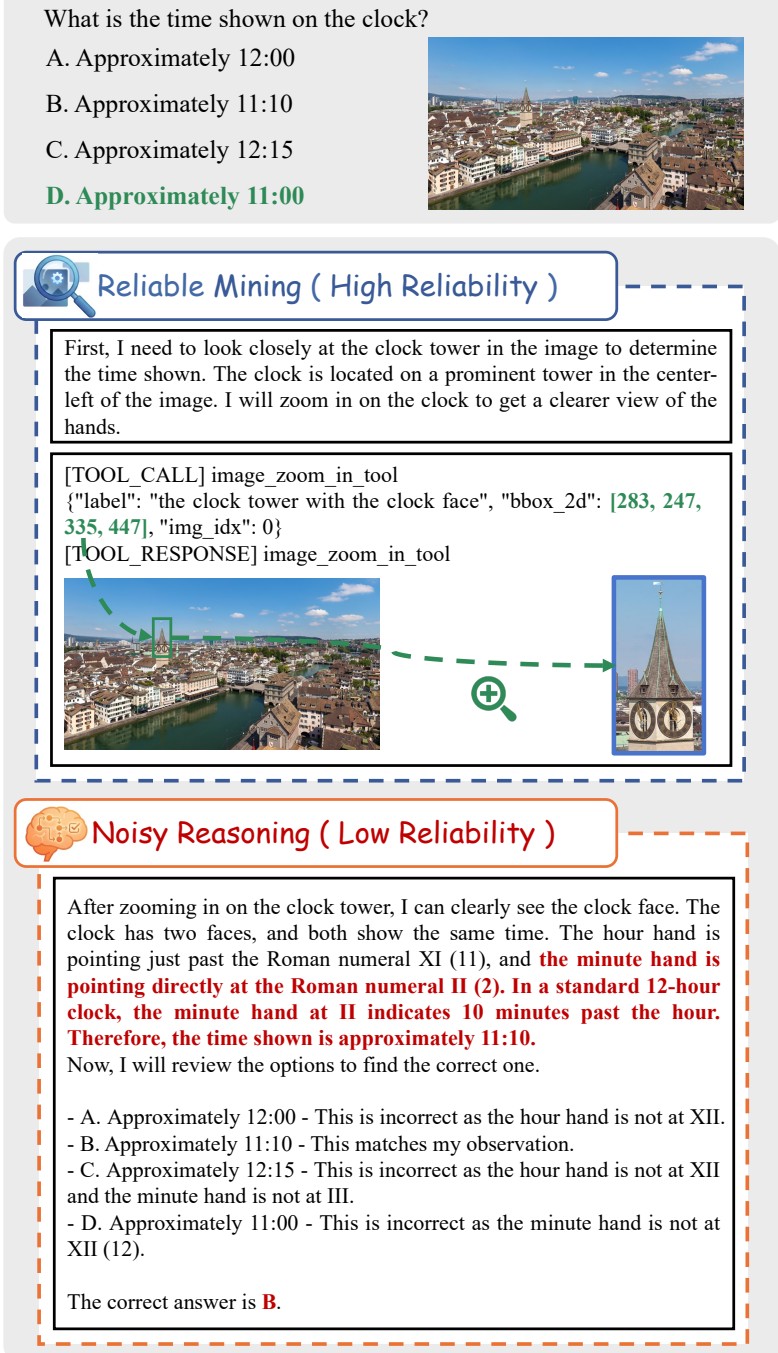

*Figure 19.* Example of noisy reasoning: symbolic misinterpretation. Although the TWI model successfully localizes the clock and zooms in on the clock face, it misinterprets the clock symbols during the reasoning stage.

