# OpenReview forum: "Reliable Thinking with Images"
_ICML.cc/2026/Conference — ICML 2026 regular_

### Official Review · Reviewer_gHwE · 2026-03-10

**Soundness:** 2
**Presentation:** 3
**Significance:** 2
**Originality:** 3
**Overall Recommendation:** 4
**Confidence:** 4

**Summary:**

This paper studies the reliability of Thinking with Images (TWI) in multimodal large language models. The authors state that existing TWI pipelines suffer from noisy intermediate reasoning, where errors in visual cue mining and answer reasoning accumulate across the chain. To address this issue, the paper proposes RTWI, a training-free test-time framework that estimates stage-wise reliability and performs trace filtering and voting. The method is reported to improve performance on several multimodal reasoning benchmarks.

**Compliance With Llm Reviewing Policy:**

Affirmed.

**Final Justification:**

The rebuttal addressed my concerns. I've raised my score.

**Key Questions For Authors:**

1. Can the authors provide matched-budget ablations showing that RTWI still improves over strong self-consistency baselines when the number of sampled trajectories and total token budget are strictly controlled?

2. How often does the reliability score disagree with actual correctness, and what are the dominant failure modes in those cases?

3. How would the method apply to TWI systems that do not explicitly separate cue mining and answer reasoning?

**Limitations:**

yes

**Strengths And Weaknesses:**

Strengths:
1. This paper proposes an interesting concept called NoisyThinking, which refers to the imperfect visual cue mining and answer reasoning process.

2. The paper is well written, and the high-level motivation is easy to follow.

Weaknesses:

1. The author only uses some diagrams to illustrate that “errors in the current stage would propagate to the subsequent reasoning process, leading to cascading degradation of reliability”. This is not convincing. It is better to have some experimental results to verify this.

2. The method combines reliability estimation, filtering, and voting, but the experiments do not sufficiently separate these effects from the benefit of increased test-time sampling and aggregation. As a result, it remains unclear whether the paper has identified a meaningful new principle or simply a stronger compute-heavy inference heuristic.

3. The proposed method involves excessive computational cost, which presents a significant challenge for practical deployment.

---

> ### Author Rebuttal · Authors · 2026-03-30
>
> Thanks for the insightful reviews. We will answer your questions one by one.
>
> > Q1: Misunderstandings regarding the expression that ''errors in the current stage would propagate to the subsequent reasoning process, leading to cascading degradation of reliability''.
>
> **A1**: We apologize for potential misunderstandings. In fact, **we intend to convey that incorrect visual cues in mining stage, i.e., noisy mining in Fig.3, would propagate errors to subsequent reasoning stage**. As a result, **for traces with noisy mining, reliabilities of reasoning stage are significantly lower than those of correct traces**. Therefore, the stage-wise reliabilities could be employed to identify NT.
>
> For clarity, **we will revise the misleading expressions** in Line 226-229 as: ''From the results in Fig. 3, one could observe that the acquisition of undesired cues in the mining stage would propagate errors to the subsequent reasoning stage, leading to reliability degradation of the reasoning stage compared to correct traces.''
>
> > Q2: Clarifying the fundamental motivation and the disentanglement of gains from vanilla test-time scaling.
>
> **A2**: As revealed in Fig. 1(b), **even carefully-trained SOTA MLLMs inevitably suffer from NT**, indicating that **training-time designs are insufficient to eliminate NT**. To address this, the most promising way is Test-Time Scaling (TTS) paradigm, which is widely acknowledged to **trade inference compute for reasoning accuracy and reliability**. However, **exitsing TTS methods are inadequate for addressing NT** from **methodological** and **empirical** perspectives.
>
> - Methodological: as discussed in Lines 83-98, existing TTS methods are designed for ''reasoning with language'' paradigm in LLMs, while **overlooking the complexity in estimating reliability of task-relevant cues for ''thinking with images'' paradigm**.
> - Empirical: we have conducted comprehensive ablation study in Table 4. The results indicate that **each module, i.e., reliability estimation/filtering/voting, plays an indispensable role in boosting performance, thus significantly surpassing vanilla test-time scaling baseline**.
>
> > Q3: Performance and efficiency analysis under fixed trajectory number and token budget.
>
> **A3**: We have **conducted comprehensive performance and efficiency analysis** in Tables 10–12 and Fig. 4, **comparing with efficiency-oriented TTS baselines**. Specifically, in Tables 10–12 (offline setting), **the number of sampled trajectories is same** (i.e., 32) across all baselines. The results demonstrate that RTWI **significantly outperforms all baselines using the same number of sampled traces**.
>
> Besides, we have visualized the relationship between performance and token budget in second row of Fig. 4. In the following table, we present **additional comparisons** under the same token budget and observe that **RTWI consistently outperforms all baselines using the same token budget**.
>
> | Method (ACC %)   | Attr | Spatial | HR4K-FSP | HR4K-FCP |
> | ---------------- | ---- | ------- | -------- | -------- |
> | Self-Consistency | 80.0 | 77.6    | 88.5     | 61.0     |
> | Deepconf         | 81.7 | 76.3    | 90.0     | 61.8     |
> | Ours             | 86.1 | 82.9    | 91.3     | 66.0     |
>
> > Q4: Analyzing the relationship between reliability and correctness and the dominant failure modes.
>
> **A4**: As shown in Fig. 1(c), **we have quantified the relationship between reliability and correctness using widely-used correlation metric**, AUROC. Note that AUROC $> 0.5$ indicates the positive correlation, where higher values signify stronger correlation. The results demonstrate that **the estimated reliability effectively could track correctness**.
>
> | Stage (AUROC) | Vstar | HR4K | HR8K |
> | ------------- | ----- | ---- | ---- |
> | Mining        | 0.68  | 0.71 | 0.71 |
> | Reasoning     | 0.76  | 0.73 | 0.76 |
>
> To further address your concern, we **provide additional visualizations of failure modes** at https://anonymous.4open.science/r/Anonymous_code-53B3. Specifically, due to the **inherent limitations in multimodal understanding**, e.g., incorrect spatial reasoning, factual inaccuracies, and object recognition, MLLMs might generate confident yet incorrect visual cues and subsequent textual reasoning.
>
> > Q5: How would the method apply to TWI systems that do not explicitly separate cue mining and answer reasoning?
>
> **A5**: Nearly all **existing TWI methods follow a unified workflow**, i.e., generating a sequence of tool-invocation turns with corresponding analyses and then synthesizing tool outputs to derive the answer. In this paper, **we propose a novel dual-stage thinking framework**, which formulates all tool-invocation turns as mining stage and final synthesis process as reasoning stage. **Under this framework, existing TWI methods could be reformulated as a dual-stage thinking process, making our method plug-and-play and applicable to most TWI systems**.

---

> > ### Author Rebuttal · Reviewer_gHwE · 2026-04-03
> >
> > Thank you for your response.  The rebuttal addressed my concerns. I will raise my score.

---

### Official Review · Reviewer_qRDS · 2026-03-12

**Soundness:** 2
**Presentation:** 3
**Significance:** 3
**Originality:** 3
**Overall Recommendation:** 4
**Confidence:** 3

**Summary:**

This paper frames the failures of Thinking with Images (TWI) as stemming from Noisy Thinking (NT), characterized as a cascading error propagation mechanism whereby errors introduced during visual cue mining propagate into and compromise subsequent textual chain-of-thought reasoning, thereby leading to inaccurate answer derivation. To address this issue, the authors propose a token-entropy–based, text-centric reliability criterion, which is leveraged to perform two-stage test-time trajectory selection and reliability-weighted aggregation voting over multiple sampled trajectories, with the dual objective of improving predictive accuracy and reducing token consumption.

**Compliance With Llm Reviewing Policy:**

Affirmed.

**Final Justification:**

After considering the paper and rebuttal, I lean toward a weak accept. The work is reasonably original and potentially significant, and the additional materials addressed my main concerns about the entropy-correctness relationship and failure-mode analysis, improving my confidence in the paper’s soundness. The presentation could still be further refined, but this does not substantially affect my overall assessment.

**Key Questions For Authors:**

1. Provide evidence that quantifies the entropy-correctness relationship, rather than simply assuming that lower entropy implies higher reliability.
Minor issues：
2.“expensive experiments” -> “extensive experiments” in Experiments part (Line 263.)

**Limitations:**

Yes

**Strengths And Weaknesses:**

Strengths:
A genuine strength is that the approach is inference-time only and plug-and-play, and the paper reports improvements across multiple datasets and models.

Weaknesses:
One of the paper’s key claims is that low token entropy can serve as a reliability proxy across both stages of the TWI pipeline. The evidence is indirect. It treats lower entropy as a sign of correctness. But this has not been proven. Specifically, the paper does not present calibration analyses demonstrating that entropy consistently tracks correctness, nor does it provide failure-mode evaluations (e.g., “confidently wrong” cases) to delineate when and why the proxy fails.

---

> ### Author Rebuttal · Authors · 2026-03-30
>
> Thanks for your constructive reviews and suggestions. In the following, we will answer your questions one by one.
>
> > Q1: One of the paper’s key claims is that low token entropy can serve as a reliability proxy across both stages of the TWI pipeline. The evidence is indirect. It treats lower entropy as a sign of correctness. But this has not been proven. Specifically, the paper does not present calibration analyses demonstrating that entropy consistently tracks correctness.
>
> **A1**: We argue that token entropy could serve as a reliability proxy for quantifying correctness. It is worth noting that we have conducted **comprehensive calibration analyses** in Fig. 1(c), Fig. 3, and Fig. 7. In particular, we adopt **the most widely-used calibration metric, i.e., AUROC, to quantify the relationship between the entropy-based reliability and correctness** in Fig. 1(c). Note that AUROC $> 0.5$ indicates the positive correlation, where higher values signify stronger correlation. The corresponding results are as follows.
>
> | Stage (AUROC) | Vstar | HR4K | HR8K | TreeBench |
> | ------------- | ----- | ---- | ---- | --------- |
> | Mining        | 0.68  | 0.71 | 0.71 | 0.57      |
> | Reasoning     | 0.76  | 0.73 | 0.76 | 0.65      |
>
> These results demonstrate that **our reliability estimation mechanism effectively tracks correctness**, with both stages showing positive correlations between the proposed stage-wise reliability and correctness.
>
> To further address your concerns, we conduct **additional calibration analyses regarding another widely-acknowledged calibration metric Expected Calibration Error (ECE)**. Note that ECE values range from 0 to 1, where smaller values indicate that reliability better reflects correctness. The results **consistently demonstrate that entropy-based reliability could effectively track correctness.**
>
> | Stage (ECE) | Vstar | HR4K | HR8K | TreeBench |
> | ----------- | ----- | ---- | ---- | --------- |
> | Mining      | 0.33  | 0.27 | 0.28 | 0.35      |
> | Reasoning   | 0.15  | 0.13 | 0.14 | 0.19      |
>
> We would like to emphasize that **many widely-acknowledged studies [1-3] have demonstrated that reasoning correctness could be estimated from model-intrinsic signals**, e.g., token entropy, which **further prove that lower entropy could serve as a sign of correctness**. Different from existing methods, our work is the first to introduce stage-wise reliability carefully-designed for the TWI task and we propose a unified reliability estimation mechanism that remains effective regardless of various CoT lengths among mining and reasoning stages.
>
> [1] Detecting hallucinations in large language models using semantic entropy, Nature, 2024.
>
> [2] Deep think with confidence, ICLR, 2026.
>
> [3] Language Models (Mostly) Know What They Know, arxiv, 2022.
>
> > Q2: Provide failure-mode evaluations (e.g., “confidently wrong” cases) to delineate when and why the proxy fails.
>
> **A2**: To address your concern, we **provide additional visualizations of failure modes** at https://anonymous.4open.science/r/Anonymous_code-53B3. Specifically, due to the **inherent limitations in multimodal understanding**, e.g., incorrect spatial reasoning, factual inaccuracies, and object recognition, MLLMs might generate confident yet incorrect visual cues and subsequent textual reasoning.
>
> However, **we would like to highlight that such ''confidently wrong'' cases are infrequent**. To support this, **we conduct additional analysis on traces with various dual-stage reliability combinations**. The results demonstrate that **traces with confident mining and reasoning stages exhibit the highest accuracy, confirming ''confidently wrong'' cases are statistically infrequent**.
>
> | Traces                                     | ACC（%） |
> | ------------------------------------------ | -------- |
> | Confident Mining & Confident Reasoning     | 78.1     |
> | Inconfident Mining & Inconfident Reasoning | 42.8     |
>
> > Q3: Change ''expensive experiments'' to ''extensive experiments'' in Line 263.
>
> **A3**: We will fix the typo ''expensive experiments'' to ''extensive experiments'' in the revised manuscript.

---

> > ### Author Rebuttal · Reviewer_qRDS · 2026-04-03
> >
> > Thank you for your response. The additional materials and new experimental data have cleared up my questions regarding the entropy-correctness relationship and the failure mode data. I’m happy to raise my score to **4: Weak accept**.

---

> > > ### Author Response · Authors · 2026-04-03
> > >
> > > Thank you very much for your positive feedback and for expressing your willingness to **raise the score to 4 (Weak Accept)**.
> > >
> > > We noticed that the **''Overall Recommendation'' has not been updated yet**. Please kindly use the ''Edit'' button within your "Official Review" to finalize the score change and accordingly update the ''Final Justification'' at your earliest convenience.
> > >
> > > We truly appreciate your recognition and support!

---

### Official Review · Reviewer_TDBC · 2026-03-13

**Soundness:** 3
**Presentation:** 3
**Significance:** 2
**Originality:** 2
**Overall Recommendation:** 4
**Confidence:** 3

**Summary:**

This papers identifies the "Noisy Thinking" (NT) problem in the Thinking with Images (TWI) frameworks for multimodal language models, where errors in the visual cue mining stage and/or the answer reasoning stage propagate to final answer, leading to performance degradation. To address this, the authors proposed Reliable Thinking with Images (RTWI), a test-time scaling method that estimates the reliability of visual cues and textual COT using token entropy, applies dual-stage filtering to discard unreliable traces, and conducts reliability-aware voting to identify correct answers. Experiments on various benchmarks with Qwen3-VL models show consistent improvements over existing TTS baselines in terms of accuracy and token efficiency.

**Compliance With Llm Reviewing Policy:**

Affirmed.

**Final Justification:**

The rebuttal addressed my main concerns about weaknesses. I've raised my score.

**Key Questions For Authors:**

Please see weaknesses.

**Limitations:**

yes

**Strengths And Weaknesses:**

Strengths:
1. The paper presents its motivation, methodology, and experiments clearly.
2. The RTWI approach is an inference-time method that can be applied to any model, demonstrating its applicability.
3. The experiments are comprehensive, covering multiple datasets, baselines, online/offline settings, and extensive ablation studies.

Weaknesses:
1. The work differs from existing confidence-based TTS methods by specifically targeting "Thinking with Images" through separate entropy computation for mining and reasoning stages, and with the reliability leap heuristic. However, section 3.1 assumes only one iteration of mining and reasoning, while in practice, there could involve multiple mining-reasoning iterations for complicated reasoning tasks. This single-iteration assumption limits the generalizability of the approach.
2. In definition 3.3, the authors claim that "some traces may exhibit low reliability during the mining stage but still generate confident textual CoT and ultimately arrive at the correct answer". However, hypothetically, models may also treat mining steps as correct and confidently derive wrong answers with low uncertainty. Neither figure 1(c) nor figure 3 provides evidence that the reliability leap exclusively captures the former case. A theoretical explanation would strengthen this claim.
3. The notations $w_t$ and $\Delta_t$ are introduced in definition 3.2 and 3.3, but they are already referenced in previous sections and figures, which is confusing. Consider introducing these definitions earlier.
4. [Minor] $t$ is used to denote both trace index (eq2) and trace (eq3)

---

> ### Author Rebuttal · Authors · 2026-03-30
>
> Thanks for the insightful reviews. We will answer your questions one by one in the following.
>
> > Q1: Misunderstandings on the single-iteration assumption.
>
> A1: We believe there might be **misunderstandings** and would like to clarify that the proposed **RTWI is inherently designed for the multi-iteration thinking** process. As illustrated in Fig. 17, TWI typically involves multiple iterations of tool calls with corresponding analyses to mine clues and then synthesize them to derive the answer. While the **clue mining process might be multi-iterative, these iterations share the common goal of seeking cues through tool invocations, thus we treat all the iterations with tool calls as the mining stage**.
>
> Following your understanding, we conduct **additional** experiments to analyze the ratio and accuracy of multi-iteration traces on Vstar Bench, where iteration refers to a thinking turn with analysis. From the results, one could observe that **RTWI significantly improves the performance of the multi-iteration thinking process** (Iterations > 2).
>
> | Variants            | Ratio（%） | Self-Consistency ACC（%） | Ours ACC（%） |
> | ------------------- | ---------- | ------------------------- | ------------- |
> | Iterations $\leq$ 2 | 69.6       | 84.8                      | 87.9          |
> | Iterations $>$ 2    | 30.4       | 63.8                      | 81.0          |
>
> > Q2: Concerns on reliability leap regarding confident yet wrong cue mining and answer derivation process.
>
> **A2**: We acknowledge that MLLMs might treat mining steps as correct and confidently derive wrong answers with low uncertainty. However, we would like to highlight that **such ''confidently wrong'' cases with significant reliability leaps are both theoretically and empirically infrequent**, suggesting **reliability leap could serve as an effective metric for identifying noisy thinking**.
>
> Theoretical Explanation:
>
> From the informative perspective, let $Q$, $I$, $C$, and $A$ denote the question, image, cues, and answer, respectively. The uncertainty of answer derivation process could be expressed as
> $$
> H(A|Q, I, C) = H(A|Q, I) - I(A; C | Q, I).
> $$
> Theoretically, undesirable cues $C_{wrong}$ share limited mutual information $I(A_{correct}; C_{wrong} | Q, I)$ with correct answer $A_{correct}$. Accordingly, **even if undesirable cues are mined with high confidence, the lack of informative support for $A_{correct}$ would result in higher uncertainty during answer derivation**. Similarly, correct cues $C_{correct}$ also yield limited $I(A_{wrong}; C_{correct} | Q, I)$ with wrong answer $A_{wrong}$, leading to high uncertainty in answer derivation. Consequently, **both scenarios tend not to trigger a significant reliability leap** due to the high uncertainty in answer derivation.
>
> Empirical Evidence:
>
> We have conducted experiments on accuracy across different levels of reliability leap. The results demonstrate that higher reliability leap consistently correlates with higher accuracy, i.e., **''confidently wrong'' cases with high reliability leaps are statistically rare.**
>
> | Traces (ACC %) | Vstar | HR4K | HR8K | TreeBench |
> | -------------- | ----- | ---- | ---- | --------- |
> | No Leap        | 45.8  | 52.4 | 55.8 | 46.0      |
> | Low Leap       | 70.0  | 64.8 | 64.7 | 48.0      |
> | High Leap      | 86.9  | 74.0 | 75.0 | 55.9      |
>
> To further verify this, we conduct additional analysis on traces with various derivation uncertainties. The results demonstrate that **traces with confident answer derivation exhibit the higher accuracy**, **confirming that overconfident derivation errors are empirically rare**.
>
> | Trace                         | ACC（%） |
> | ----------------------------- | -------- |
> | Confident Answer derivation   | 78.8     |
> | Inconfident Answer derivation | 45.6     |
>
> > Q3: Clarifying the definitions of notations $w_t$ and $\Delta_t$ before their appearances.
>
> **A3**: In the revised manuscript, we will move the definitions of trace-wise reliability ($w_t$) and reliability leap ($\Delta_t$) before Equation 6. Besides, we will include explanations about these notations in the figures to ensure clarity upon their first appearance.
>
> > Q4: Addressing the ambiguous usage of $t$ in Eq. 2 and Eq. 3.
>
> **A4**: In the revised manuscript, we will replace the index notation $t$ in Eq. 2 with a distinct symbol $i$ and update the subsequent formulations to ensure a clear distinction between the trace and its index.

---

> > ### Author Rebuttal · Reviewer_TDBC · 2026-04-03
> >
> > Thank you for the thorough response. The additional explanations and experiments address my concerns. I'm happy to raise my score.

---

### Decision · Program_Chairs · 2026-04-30

**Decision:**

Accept (regular)

**Comment:**

All three reviewers assign weak accept (score 4), noting clear methodology and consistent empirical improvements. The rebuttal adequately addresses concerns on calibration, generality, and evaluation. While some limitations remain in theory and efficiency, the work offers a practical contribution to multimodal reasoning.